# Variance Reduced Smoothed Functional REINFORCE Policy Gradient Algorithms

## Abstract

We revisit the REINFORCE policy gradient algorithm from the literature. This algorithm typically works with reward (or cost) returns obtained over episodes or trajectories. We propose a major enhancement to the basic algorithm where we estimate the policy gradient using a smoothed functional (random perturbation) gradient estimator requiring one function measurement over a perturbed parameter. Subsequently, we also propose a two-simulation counterpart of the algorithm that has lower estimator bias. Like REINFORCE, our algorithms are trajectory-based Monte-Carlo schemes and usually suffer from high variance. To handle this issue, we propose two independent enhancements to the basic scheme: (i) use the sign of the increment instead of the original (full) increment that results in smoother albeit possibly slower convergence and (ii) use clipped costs or rewards as proposed in the Proximal Policy Optimization (PPO) based scheme. We analyze the asymptotic convergence of the algorithm in the one-simulation case as well as the case where signed updates are used and briefly discuss the changes in analysis when two-simulation estimators are used. Finally, we show the results of several experiments on various Grid-World settings wherein we compare the performance of the various algorithms with REINFORCE as well as PPO and observe that both our one and two simulation SF algorithms show better performance than these algorithms. Further, the versions of these algorithms with clipped gradients and signed updates show good performance with lower variance.

**Key Words:** REINFORCE policy gradient algorithm, smoothed functional gradient estimation, one and two simulation algorithms, stochastic gradient search, stochastic shortest path Markov decision processes, signed updates, objective function clipping.

## 1 Introduction

Policy gradient methods, see Sutton & Barto (2018); Sutton et al. (1999), are a popular class of approaches in reinforcement learning (Bertsekas (2019); Meyn (2022)). Randomized policy is normally used in these approaches that is however parameterized and one updates the policy parameter along the gradient search direction. The policy gradient theorem, cf. Sutton et al. (1999); Marbach & Tsitsiklis (2001); Cao (2007), which is a fundamental result in these approaches relies on an interchange of the gradient and expectation operators and in such cases turns out to be the expectation of the gradient of noisy performance functions much like the earlier studied perturbation analysis based sensitivity approaches for simulation optimization, see Ho & Cao (1991); Chong & Ramadge (1993).

The REINFORCE algorithm, cf. Williams (1992); Sutton & Barto (2018) is a noisy gradient scheme for which the expectation of the gradient is the policy gradient, i.e., the gradient of the expected objective w.r.t. the policy parameters. The updates of the policy parameter are however obtained once after the full return on an episode has been found. Actor-critic algorithms, see Sutton & Barto (2018); Konda & Borkar (1999); Konda & Tsitsiklis (2003); Bhatnagar et al. (2009; 2007), have been presented in the literature as alternatives to the REINFORCE algorithm as they perform incremental parameter updates at every instant but do so using two-timescale stochastic approximation algorithms.

In this paper, we revisit the REINFORCE algorithm and present new algorithms for the case of episodic tasks, also referred to as the stochastic shortest path setting. Our algorithms perform parameter updates upon termination of episodes, that is when the goal or terminal states are reached. As with REINFORCE, parameter updates are performed only at instants of visit to a prescribed recurrent state, see Cao (2007); Marbach & Tsitsiklis (2001). Our first algorithm is based on a single function measurement or simulation at a perturbed parameter value where the perturbations are obtained using independent Gaussian random variates. The problem, however, is that it suffers from a large bias in the gradient estimator. We show analytically the reason for the large bias here. Subsequently, we present the two-function measurement variant of this scheme which we show has lower bias. Our algorithms rely on a diminishing sensitivity parameter sequence $\{\delta_n\}$ that appears in the denominator of an increment term in our algorithms. This can result in high variance in the iterates at least in the initial iterates. To tackle this problem, we introduce the signed analogs of these algorithms where we only consider the sign of the increment terms (the ones that multiply the learning rates in the updates). We analyze fully the asymptotic convergence of these schemes including the ones with signed updates. Subsequently, for our experiments, we also incorporate variants that use gradient clipping as with the proximal policy optimization (PPO), see Schulman et al. (2017a). A similar scheme as our first (single-measurement) algorithm is briefly presented in Bhatnagar (2023) that however does not present any analysis of convergence or experiments. Our paper, on the other hand, not only provides a detailed analysis and experiments with the one-measurement scheme, but also analyzes several other related algorithms both for their convergence as well as empirical performance.

The REINFORCE algorithm relies on an interchange between the expectation and gradient operators. In other words, it is assumed that the gradient of the expectation of the noisy performance objective equals the expectation of the gradient of the same. While such an interchange is easily justifiable in the case of finite state systems, it is not easy to justify this interchange when the number of states is infinite resulting in an infinite summation or an integral when computing the expectation. The approaches we present bypass this problem altogether by directly estimating the gradient of the expectation of the noisy objective.

Gradient estimation in our algorithm is performed using the smoothed functional (SF) technique for gradient estimation (Rubinstein, 1981; Bhatnagar & Borkar, 2003; Bhatnagar, 2007; Bhatnagar et al., 2013). The basic problem in this setting is the following: Given an objective function $J : \mathcal{R}^d \to \mathcal{R}$ such that $J(\theta) = E_\xi[h(\theta, \xi)]$, where $\theta \in \mathcal{R}^d$ is the parameter to be tuned and $\xi$ is the noise element, the goal is to find $\theta^* \in \mathcal{R}^d$ such that $J(\theta^*) = \min_{\theta \in \mathcal{R}^d} J(\theta)$. Since the objective function $J(\cdot)$ can be highly nonlinear, one often settles for a lesser goal – that of finding a local instead of a global minimum. In this setting, the Kiefer-Wolfowitz (Kiefer & Wolfowitz, 1952) finite difference estimates for the gradient of $J$ require $2d$ function measurements. This approach does not perform well in practice when $d$ is large.

Random search methods such as simultaneous perturbation stochastic approximation (SPSA) (Spall, 1992; 1997; Bhatnagar, 2005), smoothed functional (SF) (Katkovnik & Kulchitsky, 1972; Bhatnagar & Borkar, 2003; Bhatnagar, 2007) or random directions stochastic approximation (RDSA) (Kushner & Clark, 1978; Prashanth et al., 2017) typically require much less number of simulations. For instance, the gradient based algorithms in these approaches require only one or two simulations regardless of the parameter dimension $d$. A textbook treatment of random search approaches (including both gradient and Newton algorithms) for stochastic optimization is available in Bhatnagar et al. (2013).

Before we proceed further, we present the basic Markov decision process (MDP) setting and recall the REINFORCE algorithm that we consider for the episodic setting. We remark here that there are not many analyses of REINFORCE type algorithms in the literature in the episodic or stochastic shortest path setting.

## 2 The Basic MDP Setting

By a Markov decision process (MDP), we mean a controlled stochastic process $\{X_n\}$ whose evolution is governed by an associated control-valued sequence $\{Z_n\}$. It is assumed that $X_n, n \geq 0$ takes values in a set $S$ called the state-space. Let $A(s)$ be the set of feasible actions in state $s \in S$ and $A \stackrel{\triangle}{=} \cup_{s \in S} A(s)$ denote the set of all actions. When the state is say $s$ and a feasible action $a$ is chosen, the next state seen is $s'$ with a

probability $p(s'|s,a) \triangleq P(X_{n+1} = s' \mid X_n = s, Z_n = a)$, $\forall n$. Such a process satisfies the controlled Markov property, i.e., $P(X_{n+1} = s' \mid X_n, Z_n, \ldots, X_0, Z_0) = p(s' \mid X_n, Z_n)$ a.s., $\forall n \geq 0$.

By an admissible policy or simply a policy, we mean a sequence of functions $\pi = \{\mu_0, \mu_1, \mu_2, \ldots\}$, with $\mu_k : S \to A$, $k \geq 0$, such that $\mu_k(s) \in A(s)$, $\forall s \in S$. When following policy $\pi$, a decision maker selects action $\mu_k(s)$ at instant $k$, when the state is $s$. A stationary policy $\pi$ is one for which $\mu_k = \mu_l \triangleq \mu$ (a time-invariant function), $\forall k, l = 0, 1, \ldots$. Associated with any transition to a state $s'$ from a state $s$ under action $a$, is a 'single-stage' cost $g(s, a, s')$ where $g : S \times A \times S \to \mathcal{R}$ is called the cost function. The goal of the decision maker is to select actions $a_k$, $k \geq 0$ in response to the system states $s_k$, $k \geq 0$, observed one at a time, so as to minimize a long-term cost objective. We assume here that the number of states and actions is finite.

## 2.1 The Episodic or Stochastic Shortest Path Setting

We consider here the episodic or the stochastic shortest path problem where decision making terminates once a goal or terminal state is reached. We let $1, \ldots, p$ denote the set of non-terminal or regular states and $t$ be the terminal state. Thus, $S = \{1, 2, \ldots, p, t\}$ denotes the state space for this problem Bertsekas (2019).

Our basic setting here is similar to Chapter 3 of Bertsekas (2012) (see also Bertsekas (2019)), where it is assumed that under any policy there is a positive probability of hitting the goal state $t$ in at most $p$ steps starting from any initial (non-terminal) state, that would in turn signify that the problem would terminate in a finite though random amount of time.

Under a given policy $\pi$, define

$$V_\pi(s) = E_\pi \left[ \sum_{k=0}^{T} g(X_k, \mu_k(X_k), X_{k+1}) \mid X_0 = s \right], \tag{1}$$

where $T > 0$ is a finite random time at which the process enters the terminal state $t$. Here $E_\pi[\cdot]$ indicates that all actions are chosen according to policy $\pi$ depending on the system state at any instant. We assume that there is no action that is feasible in the state $t$ and so the process terminates once it reaches $t$.

Let $\Pi$ denote the set of all admissible policies. The goal here is to find the optimal value function $V^*(i)$, $i \in S$, where

$$V^*(i) = \min_{\pi \in \Pi} V_\pi(i) = V_{\pi^*}(i), \ i \in S, \tag{2}$$

with $\pi^*$ being the optimal policy. A related goal then would be to search for the optimal policy $\pi^*$. It turns out that in these problems, there exist stationary policies that are optimal, and so it is sufficient to restrict the search to the class of stationary policies.

A stationary policy $\pi$ is called a proper policy (cf. pp.174 of Bertsekas (2012)) if

$$\hat{p}_\pi \triangleq \max_{s=1,\ldots,p} P(X_p \neq t \mid X_0 = s, \pi) < 1.$$

In other words, regardless of the initial state $i$, there is a positive probability of termination after at most $p$ stages when using a proper policy $\pi$ and moreover $P(T < \infty) = 1$ under such a policy. An admissible policy (and so also a stationary policy) can be randomized as well. A randomized admissible policy or simply a randomized policy is the sequence $\psi = \{\phi_0, \phi_1, \ldots\}$ with each $\phi_i : S \to P(A)$. In other words, given a state $s$, a randomized policy would provide a distribution $\phi_i(s) = (\phi_i(s, a), a \in A(s))$ for the action to be chosen in the $i$th stage. A stationary randomized policy is one for which $\phi_j = \phi_k \triangleq \phi$, $\forall j, k = 0, 1, \ldots$. Here and in the rest of the paper, we shall assume that the policies are stationary randomized and are parameterized via a certain parameter $\theta \in C \subset \mathcal{R}^d$, a compact and convex set. We make the following assumption:

**Assumption 1** *All stationary randomized policies $\phi_\theta$ parameterized by $\theta \in C$ are proper.*

In practice, one might be able to relax this assumption (as with the model-based analysis of Bertsekas (2012)) by (a) assuming that for policies that are not proper, $V_\pi(i) = \infty$ for at least one non-terminal state $i$ and

(b) there exists a proper policy. The optimal value function satisfies the following Bellman equation: For $s = 1, \ldots, p$,

$$V^*(s) = \min_{a \in A(s)} \left( \bar{g}(s, a) + \sum_{j=1}^{p} p(j \mid s, a) V^*(j) \right), \tag{3}$$

where $\bar{g}(s, a) = \displaystyle\sum_{j=1}^{p} p(j|s, a) g(s, a, j) + p(t|s, a) g(s, a, t)$ is the expected single-stage cost in a non-terminal state $s$ when a feasible action $a$ is chosen. It can be shown, see Bertsekas (2012), that an optimal stationary proper policy exists.

## 2.2 The Policy Gradient Theorem

Policy gradient methods perform a gradient search within the prescribed class of parameterized policies. Let $\phi_\theta(s, a)$ denote the probability of selecting action $a \in A(s)$ when the state is $s \in S$ and the policy parameter is $\theta \in C$. We assume that $\phi_\theta(s, a)$ is continuously differentiable in $\theta$. A common example here is of the parameterized Boltzmann or softmax policies. Let $\phi_\theta(s) \stackrel{\triangle}{=} (\phi_\theta(s, a), a \in A(s))$, $s \in S$ and $\phi_\theta \stackrel{\triangle}{=} (\phi_\theta(s), s \in S)$.

We assume that trajectories of states and actions are available either as real data or from a simulation. Let $G_k = \displaystyle\sum_{j=k}^{T-1} g_j$ denote the sum of costs until termination (likely when a goal state is reached) on a trajectory starting from instant $k$. Note that if all actions are chosen according to a policy $\phi$, then the value and Q-value functions (under $\phi$) would be $V_\phi(s) = E_\phi[G_k \mid X_k = s]$ and $Q_\phi(s, a) = E_\phi[G_k \mid X_k = s, Z_k = a]$, respectively. In what follows, for ease of notation, we let $V_\theta \equiv V_{\phi_\theta}$ and $Q_\theta \equiv Q_{\phi_\theta}$, respectively.

The policy gradient theorem for episodic problems has the following form, cf. Chapter 13 of Sutton & Barto (2018):

$$\nabla V_\theta(s_0) = \sum_{s \in S} \mu(s) \sum_{a \in A(s)} \nabla_\theta \pi(s, a) Q_\theta(s, a), \tag{4}$$

where $\mu(s), s \in S$, is defined as $\mu(s) = \dfrac{\eta(s)}{\sum_{s' \in S} \eta(s')}$ where $\eta(s) = \displaystyle\sum_{k=0}^{\infty} p^k(s|s_0, \phi_\theta)$, $s \in S$, with $p^k(s|s_0, \phi_\theta)$ being the $k$-step transition probability of going to state $s$ from $s_0$ under the policy $\phi_\theta$. Proving the policy gradient theorem when the state-action spaces are finite is relatively straight forward (Sutton et al. (1999); Sutton & Barto (2018)). However, one would require strong regularity assumptions on the system dynamics and performance function as with infinitesimal perturbation analysis (IPA) or likelihood ratio approaches (Chong & Ramadge (1994); Ho & Cao (1991)) if the state-action spaces are either countably infinite or continuously-valued sets.

The REINFORCE algorithm (Sutton & Barto (2018); Williams (1992)) makes use of the policy gradient theorem as the latter is based on an interchange between the gradient and expectation operators (since the value function is an expectation over noisy cost returns). In what follows, we present an alternative algorithm based on REINFORCE that incorporates a one-measurement SF gradient estimator. Our algorithm does not incorporate the policy gradient theorem and thus does not require an interchange between the aforementioned operators. Our basic algorithm incorporates a zero-order gradient approximation using the smoothed functional method and like the REINFORCE algorithm, requires one sample trajectory. However, since our algorithm caters to episodic tasks, it performs updates whenever a certain prescribed recurrent state is visited, see Cao (2007); Marbach & Tsitsiklis (2001). We refer to our one-simulation algorithm as the One-SF-REINFORCE (SFR-1) algorithm.

## 3    The One-Simulation SF REINFORCE (SFR-1) Algorithm

We assume that data on the $m$th trajectory is represented in the form of the tuples $(s_k^m, a_k^m, g_k^m, s_{k+1}^m)$, $k = 0, 1, \ldots, T_m$ with $T_m$ being the termination instant on the $m$th trajectory, $m \geq 1$. Also, $s_j^m$ is the state at instant $j$ in the $m$th trajectory. Further, $a_k^m$ and $g_k^m$ are the action chosen and the cost incurred, respectively, at instant $k$ in the $m$th trajectory. Let $\Gamma : \mathcal{R}^d \to C$ denote a projection operator that projects any $x = (x_1, \ldots, x_d)^T \in \mathcal{R}^d$ to its nearest point in $C$. For ease of exposition, we assume that $C$ is a $d$-dimensional rectangle having the form $C = \prod_{i=1}^{d} [a_{i,\min}, a_{i,\max}]$, where $-\infty < a_{i,\min} < a_{i,\max} < \infty$, $\forall i = 1, \ldots, d$. Then $\Gamma(x) = (\Gamma_1(x_1), \ldots, \Gamma_d(x_d))^T$ with $\Gamma_i : \mathcal{R} \to [a_{i,\min}, a_{i,\max}]$ such that $\Gamma_i(x_i) = \min(a_{i,\max}, \max(a_{i,\min}, x_i))$, $i = 1, \ldots, d$. Also, let $\mathcal{C}(C)$ denote the space of all continuous functions from $C$ to $\mathcal{R}^d$.

In what follows, we present a procedure that incrementally updates the parameter $\theta$. Let $\theta(n)$ denote the parameter value obtained after the $n$th update of this procedure which depends on the $n$th episode and which is run using the policy parameter $\Gamma(\theta(n) + \delta_n \Delta(n))$, for $n \geq 0$, where $\theta(n) = (\theta_1(n), \ldots, \theta_d(n))^T \in \mathcal{R}^d$, $\delta_n > 0$ $\forall n$ with $\delta_n \to 0$ as $n \to \infty$ and $\Delta(n) = (\Delta_1(n), \ldots, \Delta_d(n))^T, n \geq 0$, where $\Delta_i(n), i = 1, \ldots, d, n \geq 0$ are independent random variables distributed according to the $N(0, 1)$ distribution.

Algorithm (5) below is used to update the parameter $\theta \in C \subset \mathcal{R}^d$. Let $\chi^n$ denote the $n$th state-action trajectory $\chi^n = \{s_0^n, a_0^n, s_1^n, a_1^n, \ldots, s_{T-1}^n, a_{T-1}^n, s_T^n\}$, $n \geq 0$ where the actions $a_0^n, \ldots, a_{T-1}^n$ in $\chi^n$ are obtained using the policy parameter $\theta(n) + \delta_n \Delta(n)$. The instant $T$ denotes the termination instant in the trajectory $\chi^n$ and corresponds to the instant when the terminal or goal state $t$ is reached. Note that the various actions in the trajectory $\chi^n$ are chosen according to the policy $\phi_{(\theta(n)+\delta_n\Delta(n))}$. The initial state is assumed to be sampled from a given initial distribution $\nu = (\nu(i), i \in S)$ over states. Let $G^n = \sum_{k=0}^{T-1} g_k^n$ denote the sum of costs until termination on the trajectory $\chi^n$ with $g_k^n \equiv g(s_k^n, a_k^n, s_{k+1}^n)$. The update rule that we consider here is the following: For $n \geq 0, i = 1, \ldots, d$,

$$\theta_i(n+1) = \Gamma_i\left(\theta_i(n) - a(n)\left(\Delta_i(n)\frac{G^n}{\delta_n}\right)\right). \tag{5}$$

**Assumption 2** *The step-size sequence $\{a(n)\}$ satisfies $a(n) > 0$, $\forall n$. Further,*

$$\sum_n a(n) = \infty, \ \ \sum_n \left(\frac{a(n)}{\delta_n}\right)^2 < \infty.$$

After the $(n-1)$st episode, $\theta(n)$ is computed using (5). The perturbed parameter $\theta(n) + \delta_n \Delta(n)$ is then obtained after sampling $\Delta(n)$ from the multivariate Gaussian distribution as explained previously and thereafter a new trajectory governed by this perturbed parameter is generated with the initial state in each episode sampled according to a given distribution $\nu$.

## 4    Variants for Improved Performance

We present here two variants of this algorithm that result in improved performance. The first variant reduces the bias in the estimator by using two simulations instead of one, while the second variant uses the sign of the increments instead of the increments themselves and this helps reduces the estimator variance. When applied on two-simulation SF, the second variant helps reduce both the bias and variance.

### 4.1    Two-Sided SF REINFORCE Algorithm

The idea here is to use two system simulations instead of one in order to reduce the estimator bias. As with the one-simulation SF algorithm, we assume that we have access to trajectories of data that are used for performing the parameter updates.

Let $\chi^{n+}$ and $\chi^{n-}$ denote two state-action trajectories or episodes generated after the $n$th update of the parameter. These correspond to $\chi^{n+} = \{s_0^{n+}, a_0^{n+}, s_1^{n+}, a_1^{n+}, \ldots, s_{T-1}^{n+}, a_{T-1}^{n+}, s_T^{n+}\}$, $n \geq 0$ where the actions $a_0^{n+}, \ldots, a_{T-1}^{n+}$ in $\chi^{n+}$ are obtained using the policy parameter $\theta(n) + \delta_n \Delta(n)$. Likewise, the actions $a_0^{n-}, \ldots, a_{T-1}^{n-}$ in $\chi^{n-}$ are obtained using the policy parameter $\theta(n) - \delta_n \Delta(n)$. As before, a new random vector $\Delta(n)$ is generated after $\theta(n)$ is obtained using the algorithm but the same $\Delta(n)$ is used in both the policy parameters used to generate the two trajectories.

The initial state in both these episodes is independently sampled from the same initial distribution $\nu = (\nu(i), i \in S)$ over states. Let $G^{n+} = \sum_{k=0}^{T-1} g_k^{n+}$ denote the return or the sum of costs until termination on the trajectory $\chi^{n+}$, with $g_k^{n+} \equiv g(s_k^{n+}, a_k^{n+}, s_{k+1}^{n+})$. Similarly, we let $G^{n-} = \sum_{k=0}^{T-1} g_k^{n-}$ denote the return or the sum of costs until termination on the trajectory $\chi^{n-}$, with $g_k^{n-} \equiv g(s_k^{n-}, a_k^{n-}, s_{k+1}^{n-})$.

The update rule that we consider here is the following: For $n \geq 0, i = 1, \ldots, d$,

$$\theta_i(n+1) = \Gamma_i \left( \theta_i(n) - a(n) \left( \Delta_i(n) \frac{(G^{n+} - G^{n-})}{2\delta_n} \right) \right). \tag{6}$$

### 4.2 SF REINFORCE with Signed Updates

As expected and (also) reported in the literature (Sutton & Barto (2018)), the REINFORCE algorithm typically suffers from high iterate-variance. We observe this problem even when SF-REINFORCE is used. To counter the problem of high iterate-variance, we use the sign function $sgn(\cdot)$ in the updates defined as follows: $sgn(x) = +1$ if $x > 0$ and $sgn(x) = -1$ otherwise. The update rules for the one and two simulation SF with signed updates are as follows:

#### 4.2.1 One-SF with Signed Updates

The update rule is exactly the same as (5) except that only the sign of the increment is used in the update: $\forall i = 1, \ldots, d$,

$$\theta_i(n+1) = \Gamma_i \left( \theta_i(n) - a(n) sgn \left( \Delta_i(n) \frac{G^n}{\delta_n} \right) \right). \tag{7}$$

#### 4.2.2 Two-SF with Signed Updates

As with the One-SF case, the update rule here is the same as (6) except that the update rule involves the sign of the update increment. Thus, we have $\forall i = 1, \ldots, d$,

$$\theta_i(n+1) = \Gamma_i \left( \theta_i(n) - a(n) sgn \left( \Delta_i(n) \frac{(G^{n+} - G^{n-})}{2\delta_n} \right) \right). \tag{8}$$

## 5 Convergence Analysis

We present here the main convergence results for the algorithms: one-simulation SF, two-simulation SF, and two-simulation signed SF, respectively. The proofs of all these results are provided in Appendix A.

### 5.1 Convergence of One-Simulation SF

We begin by rewriting the recursion (5) as follows:

$$\theta_i(n+1) = \Gamma_i \left( \theta_i(n) - a(n) E \left[ \Delta_i(n) \frac{G^n}{\delta_n} | \mathcal{F}_n \right] + M_{n+1}^i \right), \tag{9}$$

where $M_{n+1}^i = \Delta_i(n) \frac{G^n}{\delta_n} - E \left[ \Delta_i(n) \frac{G^n}{\delta_n} | \mathcal{F}_n \right], n \geq 0$. Here, we let $\mathcal{F}_n \triangleq \sigma(\theta(m), m \leq n, \Delta(m), \chi^m, m < n), n \geq 1$ as a sequence of increasing sigma fields with $\mathcal{F}_0 = \sigma(\theta(0))$. Let $M_n \triangleq (M_n^1, \ldots, M_n^d)^T, n \geq 0$.

**Lemma 1** $(M_n, \mathcal{F}_n), n \geq 0$ *is a martingale difference sequence.*

**Proposition 1** *We have*

$$E\left[\Delta_i(n)\frac{G^n}{\delta_n} \mid \mathcal{F}_n\right] = \sum_{s \in S} \nu(s)\nabla_i V_{\theta(n)}(s) + o(\delta_n) \ a.s.$$

In the light of Proposition 1, we can rewrite (5) as follows:

$$\theta(n+1) = \Gamma(\theta(n) - a(n)(\sum_s \nu(s)\nabla V_{\theta(n)}(s) + M_{n+1}$$

$$+\beta(n))), \tag{10}$$

where $\beta_i(n) = E\left[\Delta_i(n)\frac{G_n}{\delta} \mid \mathcal{F}_n\right] - \sum_s \nu(s)\nabla_i V_{\theta(n)}(s)$ and $\beta(n) = (\beta_1(n), \dots, \beta_d(n))^T$. From Proposition 1, it follows that $\beta(n) = o(\delta_n)$.

**Lemma 2** *The function $\nabla V_\theta(s)$ is Lipschitz continuous in $\theta$. Further, $\exists$ a constant $K_1 > 0$ such that $\| \nabla V_\theta(s) \| \leq K_1(1+ \| \theta \|)$.*

**Lemma 3** *The sequence $(M_n, \mathcal{F}_n)$, $n \geq 0$ satisfies $E[\|M_{n+1}\|^2 \mid \mathcal{F}_n] \leq \frac{\hat{L}}{\delta_n^2}$, for some constant $\hat{L} > 0$.*

Define now a sequence $Z_n, n \geq 0$ according to $Z_n = \sum_{m=0}^{n-1} a(m)M_{m+1}$, $n \geq 1$, with $Z_0 = 0$.

**Lemma 4** $(Z_n, \mathcal{F}_n)$, $n \geq 0$ *is an almost surely convergent martingale sequence.*

Consider now the following ODE:

$$\dot{\theta}(t) = \bar{\Gamma}(-\sum_s \nu(s)\nabla V_\theta(s)), \tag{11}$$

where $\bar{\Gamma} : \mathcal{C}(C) \to \mathcal{C}(\mathcal{R}^d)$ is defined according to

$$\bar{\Gamma}(v(x)) = \lim_{\eta \to 0}\left(\frac{\Gamma(x + \eta v(x)) - x}{\eta}\right). \tag{12}$$

Let $H \triangleq \{\theta \mid \bar{\Gamma}(-\sum_s \nu(s)\nabla V_\theta(s)) = 0\}$ denote the set of all equilibria of (11). By Lemma 11.1 of Borkar (2022), the only possible $\omega$-limit sets that can occur as invariant sets for the ODE (11) are subsets of $H$. Let $\bar{H} \subset H$ be the set of all internally chain recurrent points of the ODE (11). Our main result below is based on Theorem 5.3.1 of Kushner & Clark (1978) for projected stochastic approximation algorithms. We state this theorem in Appendix A along with the assumptions needed there that we verify for our analysis.

**Theorem 1** *The iterates $\theta(n), n \geq 0$ governed by (5) converge almost surely to $\bar{H}$.*

## 5.2 Convergence of Two-Simulation SF

The analysis proceeds in a similar manner here as for the one-simulation SF. Let

$$H_i(\theta(n), \Delta(n)) = \Delta_i(n)\left[\frac{V_{\theta(n)+\delta(n)\Delta(n)} - V_{\theta(n)-\delta(n)\Delta(n)}}{2\delta(n)}\right].$$

**Proposition 2**

$$E\left[\Delta_i(n)\left(\frac{G^{n+} - G^{n-}}{2\delta_n}\right) \mid \mathcal{F}_n\right] = \sum_s \nu(s)E[H_i(\theta(n), \Delta(n))|\mathcal{F}_n] = \sum_{s \in S} \nu(s)\nabla_i V_{\theta(n)}(s) + o(\delta_n) \ a.s.$$

The main result on convergence of the stochastic recursions is the following:

**Theorem 2** *The iterates $\theta(n), n \geq 0$ governed by (6) converge almost surely to $\bar{H}$.*

### 5.3 Convergence of Two-Simulation Signed SF REINFORCE

We present here the convergence analysis of the two-simulation signed SF REINFORCE algorithm. The analysis of the one-simulation counterpart is similar and hence is not provided.

Define now $e_i(n) = H_i(\theta(n), \Delta(n)) - \nabla_i V_{\theta(n)}$, and let $F_i(e|\theta) = P(e_i(n) \leq e|\theta(n) = \theta)$ be the conditional distribution of $e_i(n)$ given $\theta(n) = \theta$. We make the following assumptions:

(A1) $P(e_i(n) \leq e|\theta(m), m \leq n) = F_i(e|\theta(n))$ independent of $n$.

(A2) The maps $(e, \theta) \mapsto F_i(e|\theta)$ and $\theta \mapsto \nabla_i V_\theta$ are Lipschitz continuous.

(A3) For all $\theta$ and $i = 1, \ldots, N$, $F_i(0|\theta) = 1/2$.

(A4) $a(n) > 0, \forall n, \sum_n a(n) = \infty, \sum_n \left( \dfrac{a(n)}{\delta(n)} \right)^2 < \infty.$

Consider the following ODE associated with the above recursion:

$$\dot{\theta}_i(t) = \bar{\Gamma}_i(-(1 - 2F_i(-\sum_s \nu(s)\nabla_i V_\theta(s)|\theta))), \ t \geq 0, \ i = 1, \ldots, N. \tag{13}$$

For $x = (x_1, \ldots, x_d)^T$, let $\bar{\Gamma}(x) = (\bar{\Gamma}_1(x_1), \ldots, \bar{\Gamma}_d(x_d))^T$. Also, let $F(-\nabla V_\theta)|\theta) \stackrel{\triangle}{=} (F_1(-\sum_s \nu(s)\nabla_1 V_\theta(s)|\theta),$ $\ldots, F_d(-\sum_s \nu(s)\nabla_N V_\theta(s)|\theta))$ and let $K = \{\theta|(\bar{\Gamma}(-(1 - 2F(-\sum_s \nu(s)\nabla V_\theta(s)|\theta)) = 0)\}$ denote the set of equilibria of (13). Further, let $\bar{K} \subset K \subset \{\theta|\bar{\Gamma}(\langle(1 - 2F(-\sum_s \nu(s)\nabla V_\theta(s)|\theta)), \sum_s \nu(s)\nabla V_\theta(s)\rangle) = 0\}$ denote the largest invariant set contained in $K$.

**Theorem 3 (Convergence of Signed SFR-2)** *$\{\theta(n)\}$ governed as per (8) converges as $n \to \infty$ almost surely to $\bar{K}$.*

**Remark 1** *Suppose $\theta \in K$ is such that $\theta$ is in the interior of the constraint set. Then, from Assumptions (A2)-(A3) and Theorem 3, $\sum_s \nu(s)\nabla V_\theta(s) = 0$. For $\theta$ on the boundary of the constraint set, either $\sum_s \nu(s)\nabla V_\theta(s) = 0$ or $\sum_s \nu(s)\nabla V_\theta(s) \neq 0$ but in the latter case, $\bar{\Gamma}(\sum_s \nu(s)\nabla V_\theta(s) = 0)$. The latter are spurious fixed points that occur at the boundary of the constraint set, see Kushner & Yin (1997).*

## 6 Numerical Results

We investigate the performance of our proposed SF-REINFORCE algorithms (both one and two sided variants: SFR-1 and SFR-2) on a stochastic grid world setting. The specifics of the environment setup, policy estimator and the algorithm parameters are discussed in Appendix B.1, B.2 and B.3, respectively.

Initially, we experiment with various grid sizes and compare our SFR-1 and SFR-2 with REINFORCE and PPO. To ensure the results are reproducible, we run every setting 10 times with different seeds, and we plot the mean as the dark line, and shade using the standard deviation around it. Although our algorithms show good performance, they also display large variance. We refer the reader to Appendix B.4 for these experiments. We further try out various schedules for perturbations and learning rates, as well as gradient clipping techniques. In the process of trying out methods to improve on the variance of the objective, we illustrate the tradeoff between the performance of the iterates and their variance.

For brevity, we show the best performance on our largest (most difficult) grid in Figure 1 and also compare how fast these algorithms achieve a performance threshold, by counting the number of updates in Table 2. Table 1 summarizes the best performance of variants of SFR-1 and SFR-2 on $50 \times 50$ grid. Notably, SFR-2

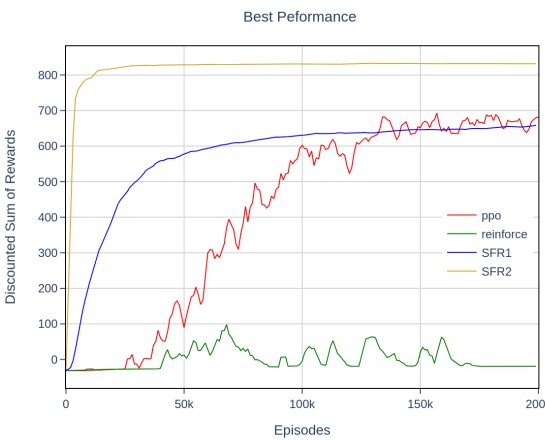

Figure 1: Best runs across all algorithms on $50 \times 50$ grid

| algo | vanilla | const_delta | signed | clip_by_value | clip_by_norm |
|------|---------|-------------|--------|----------------|---------------|
| SFR-1 | $376.29 \pm 401.4$ | $\mathbf{677.55 \pm 233.1}$ | $424.93 \pm 254.5$ | $483.68 \pm 215.3$ | $395.38 \pm 279.6$ |
| SFR-2 | $562.89 \pm 387.7$ | $665.5 \pm 229.1$ | $823.68 \pm 3.1$ | $824.96 \pm 5.6$ | $\mathbf{830.4 \pm 3.7}$ |

Table 1: Best Performance of variants, on large grid, $L = 50$

with gradient clipping by norm and SFR-1 with constant perturbation size emerge as the leaders in their respective best runs. Both algorithms require substantially fewer updates compared to PPO. While neither SFR-1 nor PPO reach thresholds of 700 and 800 (resp.), SFR-2 achieves a target of 800 within just 11,000 updates. This demonstrates the superior efficacy of SFR-2, particularly when incorporating techniques like gradient clipping.

## 6.1 Perturbation sizes

Since our algorithm uses stochastic perturbations to estimate the gradient of the objective function, we expect the dynamics of our policy weights to be intimately related to the perturbation size. To investigate this, we experiment with both decaying and constant perturbation schedules $\delta(n)$.

### 6.1.1 Decaying perturbations

Since we are using $\delta(n) = \delta_0(\frac{1}{50000+n})^d$, we set $\delta_0 = 1$, $\alpha_0 = 2 \times 10^{-6}$ and vary $d \in \{0.15, 0.25, 0.35, 0.45\}$. These runs are made on the medium grid size $L = 10$. Note that for a fixed $\delta_0$, lower values of $d$ correspond to higher values of $\delta(n)$. It is clear that for larger perturbations ($d \in \{0.101, 0.15, 0.25\}$), there is lower variance, both across runs (in terms of the converged value) and lower variance within the run (compare the shaky blue lines from Figures 2c and 2d with the steady lines from Figures 2a and 2b). It is clear from Figure 2 that we can conclude that SFR-1 is more sensitive to decay parameter since it affects the variance in the iterates within the same run too.

| Algorithm / Threshold | 200 | 400 | 600 | 700 | 800 |
|----------------------|-----|-----|-----|-----|-----|
| SFR-2 | 1000 | 1000 | 2000 | 3000 | 11000 |
| SFR-1 | 9000 | 19000 | 63000 | - | - |
| PPO | 51000 | 68000 | 99000 | 134000 | - |

Table 2: Number of updates from different algorithms required to cross reward threshold on $50 \times 50$ grid

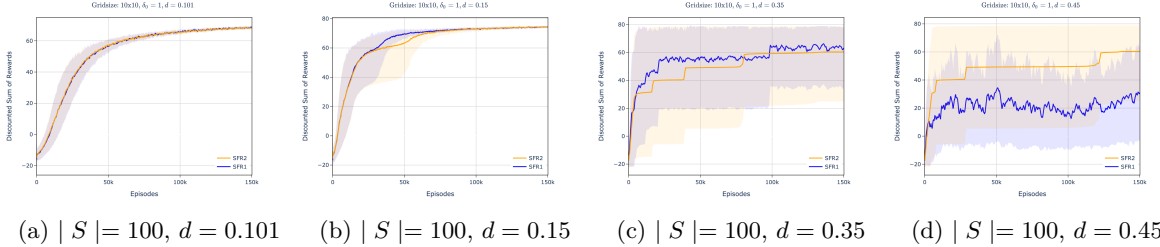

Figure 2: Iterate performance for different decay exponents $d$ for perturbations $\delta(n)$ on 10×10 grid. Other parameters are set to $\delta_0 = 1$, $\alpha_0 = 2 \times 10^{-6}$.

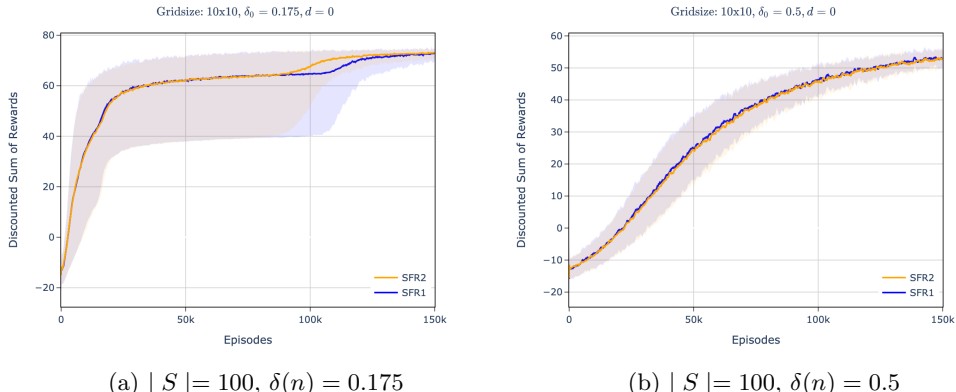

Figure 3: Runs showing constant schemes for $\delta(n)$, for grid size $L = 10$.

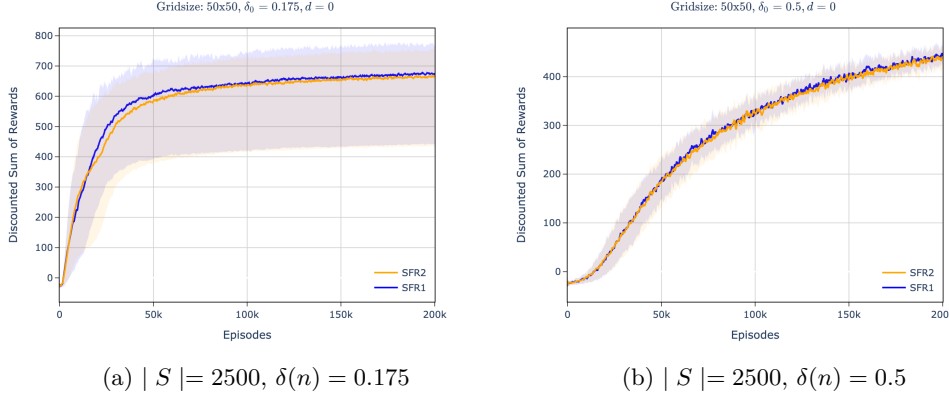

Figure 4: Runs showing constant schemes for $\delta(n)$, for the large grid size $L = 50$, capturing the tradeoff between performance and variance.

| Algorithm | $d = 0.101$ | $d = 0.15$ | $d = 0.25$ | $d = 0.35$ | $d = 0.45$ |
|---|---|---|---|---|---|
| SFR-1 | $68.72 \pm 1.5$ | $74.07 \pm 0.7$ | $59.45 \pm 35.0$ | $62.60 \pm 28.4$ | $31.33 \pm 34.0$ |
| SFR-2 | $68.10 \pm 1.3$ | $74.18 \pm 0.6$ | $68.49 \pm 26.3$ | $60.40 \pm 35.4$ | $60.46 \pm 35.4$ |

Table 3: Performance for different decay exponents $d$ for perturbations $\delta(n)$ on 10×10 grid. Other parameters are set to $\delta_0 = 1$, $\alpha_0 = 2 \times 10^{-6}$.

| Algorithm / Decay | 0.175 | 0.500 |
|---|---|---|
| SFR-1 | $72.66 \pm 3.6$ | $52.60 \pm 2.8$ |
| SFR-2 | $73.29 \pm 1.9$ | $53.10 \pm 3.2$ |

Table 4: Performance for different constant values of $\delta(n)$ on $10 \times 10$ grid over 10 seeded runs

| Algorithm | $\delta(n) = 0.175$ | $\delta(n) = 0.5$ | $\delta(n) = 0.7$ |
|---|---|---|---|
| SFR-1 | $677.55 \pm 233.1$ | $446.65 \pm 26.8$ | $254.74 \pm 35.6$ |
| SFR-2 | $665.50 \pm 229.1$ | $437.26 \pm 21.7$ | $247.09 \pm 33.0$ |

Table 5: Performance for different constant values of $\delta(n)$ on $50 \times 50$ grid over 10 seeded runs

### 6.1.2 Constant Perturbations

We also tried experiments with constant values of $\delta(n) = 0.175$ and $\delta(n) = 0.5$ (they correspond to $d = 0$ and $\delta_0 \in \{0.175, 0.5\}$). We run on both medium and large grids ($L \in \{10, 50\}$). For both grids, SFR-1 and SFR-2 show similar results, see Figures 3 and 4. Larger value of $\delta(n)$ shows stable iterates that however do not reach a high enough reward compared to those with a smaller value of $\delta(n)$. Tables 4 and 5 capture the trade-off between the mean and variance of the converged values of this setting. Since $\delta(n) = 0.175$ gives the best performance on the large grid-size, we try experimenting with signed and clipped gradients to improve on the consistency of the total reward across runs.

## 6.2 On transforming updates

To control the dynamics of the weights, we employ variants of the proposed scheme geared towards transforming the gradients to mitigate the effects of exploding or vanishing gradients problem. The effect of signed as well as clipped gradients is investigated.

### 6.2.1 Signed Updates

We experiment with the signed update scheme described and analyzed previously. For this, we maintain a constant value for $\delta(n) = 0.175$, while varying the step-sizes ($\alpha(n)$) as shown in Figure 5 and Table 6. Since each component is changed by a magnitude of $\alpha(n)$, the step size controls how far the iterates can go in the span of fixed iterations, from a given random initialization. As expected, for very small step-sizes, the performance is not good, but improves with increasing step-sizes. We note that for large step-sizes, the two-sided variant does extremely well and shows good reward performance with low standard deviation as well. In the case of the one-sided algorithm, large step-sizes also improve the discounted reward performance but at the same time increase variance.

### 6.2.2 Gradient clipping

To maintain stability of the iterates, we need to ensure that gradients do not explode. One way is to clip each component of the gradient by value, whereas the other method involves normalizing the gradient such that its norm does not exceed a previously agreed value. Figure 6 shows the best runs in both cases. Results of more detailed experiments on different gridsizes are provided in Appendix B.4.

| Algorithm | $\alpha_0 = 2 \times 10^{-6}$ | $\alpha_0 = 2 \times 10^{-4}$ | $\alpha_0 = 2 \times 10^{-3}$ | $\alpha_0 = 2 \times 10^{-2}$ |
|---|---|---|---|---|
| SFR-1 | $-26.03 \pm 8.2$ | $141.32 \pm 144.6$ | $392.66 \pm 234.3$ | $424.93 \pm 254.5$ |
| SFR-2 | $-22.87 \pm 6.4$ | $456.54 \pm 161.1$ | $761.28 \pm 6.7$ | $823.68 \pm 3.1$ |

Table 6: Performance of signed updates on large grids by varying step-sizes

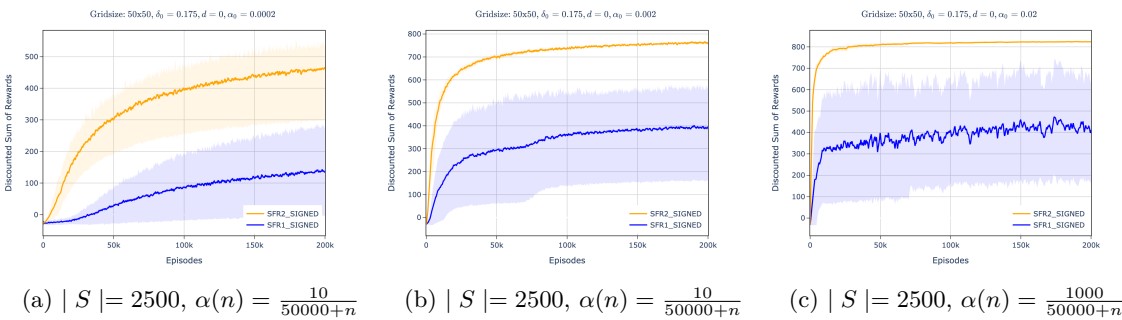

(a) $\mid S \mid= 2500, \alpha(n) = \frac{10}{50000+n}$     (b) $\mid S \mid= 2500, \alpha(n) = \frac{10}{50000+n}$     (c) $\mid S \mid= 2500, \alpha(n) = \frac{1000}{50000+n}$

Figure 5: Runs for signed updates with different values of step-sizes, $\alpha(n)$

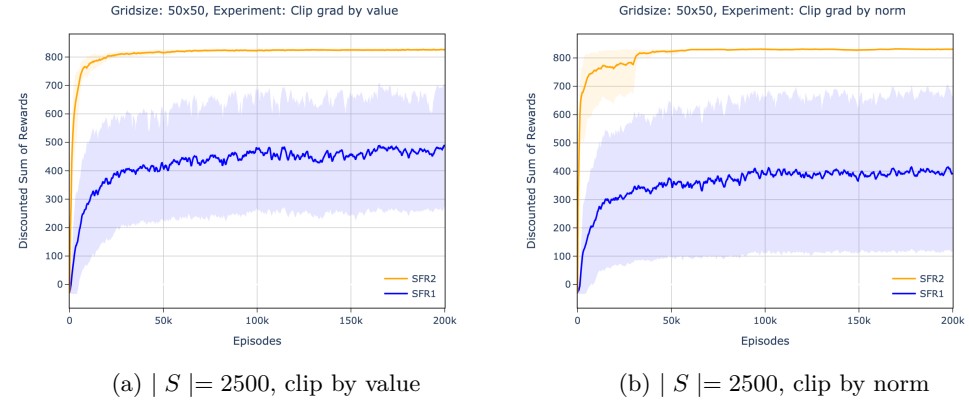

(a) $\mid S \mid= 2500$, clip by value       (b) $\mid S \mid= 2500$, clip by norm

Figure 6: Runs for gradients clipped by value and norm

## 7 Conclusions

We presented model-free smoothed functional algorithms as suitable Monte-Carlo based alternatives to the popular REINFORCE algorithm for the setting of episodic tasks. We first presented the one-simulation SF REINFORCE algorithm followed by its two-simulation counterpart. We also presented the signed variant of these algorithms. Subsequently, we provided a complete convergence analysis of the one-simulation SF REINFORCE algorithm, and described the changes in the analysis needed for the two-simulation variant. Next, we also provided the convergence analysis of the two-simulation signed variant, the one with one-simulation being analogous.

We showed detailed empirical results of the various algorithms on different grid world settings and under various choices of the setting parameters. We also compared the performance of our algorithms with the REINFORCE algorithm as well as PPO, and observed that both one-sided and two-sided SF REINFORCE algorithms show better performance than these other algorithms. We also found that signed updates and gradient clipping are effective procedures that help significantly alleviate the problems of high variance in Monte-Carlo algorithms such as REINFORCE. As future work, it would be of interest to theoretically study the convergence rate results of the algorithms presented here.

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

## A  Details of the Convergence Analysis

We present here the details of the convergence analysis and give the proofs of the various results. We begin first with the results for the One-Simulation SF algorithm. We will subsequently sketch the analysis of the two-simulation SF algorithm. Finally, we shall discuss the convergence analysis of the algorithms with signed updates.

### A.1  Convergence of One-Simulation SF

**Proof of Lemma 1:** Notice that

$$M_n^i = \Delta_i(n-1)\frac{G^{n-1}}{\delta_{n-1}} - E\left[\Delta_i(n-1)\frac{G^{n-1}}{\delta_{n-1}} \mid \mathcal{F}_{n-1}\right].$$

The first term on the RHS above is clearly measurable $\mathcal{F}_n$ while the second term is measurable $\mathcal{F}_{n-1}$ and hence measurable $\mathcal{F}_n$ as well. Further, from Assumption 1, each $M_n$ is integrable. Finally, it is easy to verify that

$$E[M_{n+1}^i \mid \mathcal{F}_n] = 0, \ \forall i.$$

The claim follows.

**Proof of Proposition 1:** Note that

$$E\left[\Delta_i(n)\frac{G^n}{\delta_n} \mid \mathcal{F}_n\right] = E\left[E\left[\Delta_i(n)\frac{G^n}{\delta_n} \mid \mathcal{G}_n\right] \mid \mathcal{F}_n\right],$$

where $\mathcal{G}_n \overset{\triangle}{=} \sigma(\theta(m), \Delta(m), m \leq n, \chi^m, m < n), n \geq 1$ is a sequence of increasing sigma fields with $\mathcal{G}_0 = \sigma(\theta(0), \Delta(0))$. It is clear that $\mathcal{F}_n \subset \mathcal{G}_n, \forall n \geq 0$. Now,

$$E\left[\Delta_i(n)\frac{G^n}{\delta_n} \mid \mathcal{G}_n\right] = \frac{\Delta_i(n)}{\delta_n}E[G^n \mid \mathcal{G}_n].$$

Let $s_0^n = s$ denote the initial state in the trajectory $\chi^n$. Recall that the initial state $s$ is chosen randomly from the distribution $\nu$. Thus,

$$E[G^n \mid \mathcal{G}_n] = \sum_s \nu(s) E[G^n \mid s_0^n = s, \phi_{\theta(n)+\delta_n \Delta(n)}]$$

$$= \sum_s \nu(s) V_{\theta(n)+\delta_n \Delta(n)}(s).$$

Thus, with probability one,

$$E\left[\Delta_i(n)\frac{G^n}{\delta_n} \mid \mathcal{G}_n\right] = \sum_s \nu(s)\left(\Delta_i(n)\frac{V_{\theta(n)+\delta_n \Delta(n)}(s)}{\delta_n}\right).$$

Hence, it follows almost surely that

$$E\left[\Delta_i(n)\frac{G^n}{\delta_n} \mid \mathcal{F}_n\right] = \sum_s \nu(s) E\left[\Delta_i(n)\frac{V_{\theta(n)+\delta_n \Delta(n)}(s)}{\delta_n} \mid \mathcal{F}_n\right].$$

Using a Taylor's expansion of $V_{\theta(n)+\delta_n \Delta(n)}(s)$ around $\theta(n)$ gives us

$$V_{\theta(n)+\delta_n \Delta(n)}(s_n) = V_{\theta(n)}(s_n) + \delta_n \Delta(n)^T \nabla V_{\theta(n)}(s_n)$$

$$+ \frac{\delta_n^2}{2}\Delta(n)^T \nabla^2 V_{\theta(n)}(s_n)\Delta(n) + o(\delta_n^2).$$

Now recall that $\Delta(n) = (\Delta_i(n), i = 1, \ldots, d)^T$. Thus,

$$\Delta(n)\frac{V_{\theta(n)+\delta_n \Delta(n)}(s_n)}{\delta_n} = \frac{1}{\delta_n}\Delta(n)V_{\theta(n)}(s_n)$$

$$+ \Delta(n)\Delta(n)^T \nabla V_{\theta(n)}(s_n)$$

$$+ \frac{\delta_n}{2}\Delta(n)\Delta(n)^T \nabla^2 V_{\theta(n)}(s_n)\Delta(n) + o(\delta_n).$$

Now observe from the properties of $\Delta_i(n), \forall i, n$, that
(i) $E[\Delta(n)] = 0$ (the zero-vector), $\forall n$, since $\Delta_i(n) \sim N(0,1)$, $\forall i, n$.
(ii) $E[\Delta(n)\Delta(n)^T] = I$ (the identity matrix), $\forall n$.
(iii) $E\left[\sum_{i,j,k=1}^{d} \Delta_i(n)\Delta_j(n)\Delta_k(n)\right] = 0.$

Property (iii) follows from the facts that (a) $E[\Delta_i(n)\Delta_j(n)\Delta_k(n)] = 0$, $\forall i \neq j \neq k$, (b) $E[\Delta_i(n)\Delta_j^2(n)] = 0$, $\forall i \neq j$ (this pertains to the case where $i \neq j$ but $j = k$ above) and (c) $E[\Delta_i^3(n)] = 0$ (for the case when $i = j = k$ above). These properties follow from the independence of the random variables $\Delta_i(n)$, $i = 1, \ldots, d$ and $n \geq 0$, as well as the fact that they are all distributed $N(0,1)$. The claim now follows from (i)-(iii) above.

Recall that from Proposition 1, it follows that $\beta(n) = o(\delta_n)$.

**Proof of Lemma 2**: It can be seen from (4) that $V_\theta(s)$ is continuously differentiable in $\theta$. It can also be shown as in Theorem 3 of Furmston et al. (2016) that $\nabla^2 V_\theta(s)$ exists and is continuous. Since $\theta$ takes values in $C$, a compact set, it follows that $\nabla^2 V_\theta(s)$ is bounded and thus $\nabla V_\theta(s)$ is Lipschitz continuous.

Finally, let $L_1^s > 0$ denote the Lipschitz constant for the function $\nabla V_\theta(s)$. Then, for a given $\theta_0 \in C$,

$$\| \nabla V_\theta(s) \| - \| \nabla V_{\theta_0}(s) \| \leq \| \nabla V_\theta(s) - \nabla V_{\theta_0}(s) \|$$

$$\leq L_1^s \| \theta - \theta_0 \|$$

$$\leq L_1^s \parallel \theta \parallel + L_1^s \parallel \theta_0 \parallel.$$

Thus, $\parallel \nabla V_\theta(s) \parallel \leq \parallel \nabla V_{\theta_0}(s) \parallel + L_1^s \parallel \theta_0 \parallel + L_1^s \parallel \theta \parallel$. Let $K_s \triangleq \parallel \nabla V_{\theta_0}(s) \parallel + L_1^s \parallel \theta_0 \parallel$ and $K_1 \triangleq \max(K_s, L_1^s, s \in S)$. Thus, $\parallel \nabla V_\theta(s) \parallel \leq K_1(1 + \parallel \theta \parallel)$. Note here that since $|S| < \infty$, $K_1 < \infty$ as well. The claim follows.

**Proof of Lemma 3:** Note that

$$\|M_{n+1}\|^2 = \sum_{i=1}^{d}(M_{n+1}^i)^2$$

$$= \sum_{i=1}^{d}\left(\Delta_i^2(n)\frac{(G^n)^2}{\delta_n^2} + \frac{1}{\delta_n^2}E\left[\Delta_i(n)G^n \mid \mathcal{F}_n\right]^2\right.$$

$$\left.-2\Delta_i(n)\frac{G^n}{\delta_n^2}E\left[\Delta_i(n)G^n \mid \mathcal{F}_n\right]\right).$$

Thus,

$$E[\|M_{n+1}\|^2 \mid \mathcal{F}_n] = \frac{1}{\delta_n^2}\sum_{i=1}^{d}\left(E[\Delta_i^2(n)(G^n)^2 \mid \mathcal{F}_n]\right.$$

$$\left.-E^2[\Delta_i(n)G^n \mid \mathcal{F}_n]\right).$$

The claim now follows from Assumption 1 and the fact that all single-stage costs are bounded (cf. pp.174, Chapter 3 of Bertsekas (2012)).

**Proof of Lemma 4:** It is easy to see that $Z_n$ is $\mathcal{F}_n$-measurable $\forall n$. Further, it is integrable for each $n$ and moreover $E[Z_{n+1} \mid \mathcal{F}_n] = Z_n$ almost surely since $(M_{n+1}, \mathcal{F}_n)$, $n \geq 0$ is a martingale difference sequence by Lemma 1. It is also square integrable from Lemma 3. The quadratic variation process of this martingale will be convergent almost surely if

$$\sum_{n=0}^{\infty}E[\|Z_{n+1} - Z_n\|^2 \mid \mathcal{F}_n] < \infty \text{ a.s.} \tag{14}$$

Note that

$$E[\|Z_{n+1} - Z_n\|^2 \mid \mathcal{F}_n] = a(n)^2 E[\|M_{n+1}\|^2 \mid \mathcal{F}_n].$$

Thus,

$$\sum_{n=0}^{\infty}E[\|Z_{n+1} - Z_n\|^2 \mid \mathcal{F}_n] = \sum_{n=0}^{\infty}a(n)^2 E[\|M_{n+1}\|^2 \mid \mathcal{F}_n]$$

$$\leq \hat{L}\sum_{n=0}^{\infty}\left(\frac{a(n)}{\delta_n}\right)^2,$$

by Lemma 3. (14) now follows as a consequence of Assumption 2. Now $(Z_n, \mathcal{F}_n)$, $n \geq 0$ can be seen to be convergent from the martingale convergence theorem for square integrable martingales Borkar (1995).

Our main result below is based on Theorem 5.3.1 of Kushner & Clark (1978) for projected stochastic approximation algorithms. Before we proceed further, we recall that result below.

Let $C \subset \mathcal{R}^d$ be a compact and convex set as before and $\Gamma : \mathcal{R}^d \to C$ denote the projection operator that projects any $x = (x_1, \ldots, x_d)^T \in \mathcal{R}^d$ to its nearest point in $C$.

Consider now the following the $d$-dimensional stochastic recursion

$$X_{n+1} = \Gamma(X_n + a(n)(h(X_n) + \xi_n + \beta_n)), \tag{15}$$

under the assumptions listed below. Also, consider the following ODE associated with (15):

$$\dot{X}(t) = \bar{\Gamma}(h(X(t))). \tag{16}$$

Let $\mathcal{C}(C)$ denote the space of all continuous functions from $C$ to $\mathcal{R}^d$. The operator $\bar{\Gamma} : \mathcal{C}(C) \to \mathcal{C}(\mathcal{R}^d)$ is defined according to

$$\bar{\Gamma}(v(x)) = \lim_{\eta \to 0} \left( \frac{\Gamma(x + \eta v(x)) - x}{\eta} \right), \tag{17}$$

for any continuous $v : C \to \mathcal{R}^d$. The limit in (17) exists and is unique since $C$ is a convex set. In case this limit is not unique, one may consider the set of all limit points of (17). Note also that from its definition, $\bar{\Gamma}(v(x)) = v(x)$ if $x \in C^o$ (the interior of $C$). This is because for such an $x$, one can find $\eta > 0$ sufficiently small so that $x + \eta v(x) \in C^o$ as well and hence $\Gamma(x + \eta v(x)) = x + \eta v(x)$. On the other hand, if $x \in \partial C$ (the boundary of $C$) is such that $x + \eta v(x) \notin C$, for any small $\eta > 0$, then $\bar{\Gamma}(v(x))$ is the projection of $v(x)$ to the tangent space of $\partial C$ at $x$.

Consider now the assumptions listed below.

(B1) The function $h : \mathcal{R}^d \to \mathcal{R}^d$ is continuous.

(B2) The step-sizes $a(n), n \geq 0$ satisfy

$$a(n) > 0 \forall n, \ \sum_n a(n) = \infty, \ a(n) \to 0 \text{ as } n \to \infty.$$

(B3) The sequence $\beta_n, n \geq 0$ is a bounded random sequence with $\beta_n \to 0$ almost surely as $n \to \infty$.

(B4) There exists $T > 0$ such that $\forall \epsilon > 0$,

$$\lim_{n \to \infty} P \left( \sup_{j \geq n} \max_{t \leq T} \left| \sum_{i=m(jT)}^{m(jT+t)-1} a(i)\xi_i \right| \geq \epsilon \right) = 0.$$

(B5) The ODE (16) has a compact subset $K$ of $\mathcal{R}^N$ as its set of asymptotically stable equilibrium points.

Let $t(n), n \geq 0$ be a sequence of positive real numbers defined according to $t(0) = 0$ and for $n \geq 1$, $t(n) = \sum_{j=0}^{n-1} a(j)$. By Assumption (B2), $t(n) \to \infty$ as $n \to \infty$. Let $m(t) = \max\{n \mid t(n) \leq t\}$. Thus, $m(t) \to \infty$ as $t \to \infty$. Assumptions (B1)-(B3) correspond to A5.1.3-A5.1.5 of Kushner & Clark (1978) while (B4)-(B5) correspond to A5.3.1-A5.3.2 there.

(Kushner & Clark, 1978, Theorem 5.3.1 (pp. 191-196)) essentially says the following:

**Theorem 4 (Kushner and Clark Theorem:)** *Under Assumptions (B1)–(B5), almost surely, $X_n \to K$ as $n \to \infty$.*

Finally, we come to the proof of our main result.

**Proof of Theorem 1:** In lieu of the foregoing, we rewrite (5) according to

$$\theta_i(n+1) = \Gamma_i \Big( \theta_i(n) - a(n) \sum_s \nu(s) \nabla_i V_{\theta(n)}(s)$$

$$- a(n)\beta_i(n) + M_{n+1}^i \Big), \tag{18}$$

where $\beta_i(n)$ is as in (10). We shall proceed by verifying Assumptions (B1)-(B5) and subsequently appeal to Theorem 5.3.1 of Kushner & Clark (1978) (i.e., Theorem 1 above) to claim convergence of the scheme. Note that Lemma 2 ensures Lipschitz continuity of $\nabla V_\theta(s)$ implying (B1). Next, from (B2), since $\delta_n \to 0$, it follows that $a(n) \to 0$ as $n \to \infty$. Thus, Assumption (B2) holds as well. Now from Lemma 2, it follows that $\sum_s \nu(s) \nabla V_\theta(s)$ is uniformly bounded since $\theta \in C$, a compact set. Assumption (B3) is now verified from Proposition 1. Since $C$ is a convex and compact set, Assumption (B4) holds trivially. Finally, Assumption

(B5) is also easy to see as a consequence of Lemma 4. Now note that for the ODE (11), $F(\theta) = \sum_s \nu(s)V_\theta(s)$ serves as an associated Lyapunov function and in fact

$$\nabla F(\theta)^T \bar{\Gamma}(-\sum_s \nu(s)\nabla V_\theta(s))$$

$$= (\sum_s \nu(s)\nabla_\theta V_\theta(s))^T \bar{\Gamma}(-\sum_s \nu(s)\nabla V_\theta(s)) \le 0.$$

For $\theta \in C^o$ (the interior of $C$), it is easy to see that $\bar{\Gamma}(-\sum_s \nu(s)\nabla V_\theta(s)) = -\sum_s \nu(s)\nabla V_\theta(s)$, and

$$\nabla F(\theta)^T \bar{\Gamma}(-\sum_s \nu(s)\nabla V_\theta(s)) \quad < \quad 0 \text{ if } \theta \in H^c \cap C$$

$$= \quad 0 \text{ o.w.}$$

For $\theta \in \delta C$ (the boundary of $C$), there can additionally be spurious attractors, see Kushner & Yin (1997), that are also contained in $H$. The claim now follows from Theorem 5.3.1 of Kushner & Clark (1978).

## A.2 Convergence of Two-Simulation SF

The analysis proceeds in a similar manner as for the one-simulation SF except with $\dfrac{G^{n+} - G^{n-}}{2\delta_n}$ in place of $\dfrac{G^n}{\delta_n}$.

**Proof of Proposition 2:**

A similar calculation as with the proof of Proposition 1 would show that

$$E\left[\Delta_i(n)\left(\frac{G^{n+} - G^{n-}}{2\delta_n}\right) \mid \mathcal{F}_n\right] = \sum_s \nu(s)E\left[\Delta_i(n)\frac{(V_{\theta(n)+\delta_n\Delta(n)}(s) - V_{\theta(n)-\delta_n\Delta(n)}(s)}{2\delta_n} \mid \mathcal{F}_n\right].$$

Using Taylor's expansions of $V_{\theta(n)+\delta_n\Delta(n)}(s)$ and $V_{\theta(n)-\delta_n\Delta(n)}(s)$ around $\theta(n)$ gives us

$$\Delta(n)\left(\frac{V_{\theta(n)+\delta_n\Delta(n)}(s_n) - V_{\theta(n)-\delta_n\Delta(n)}(s_n)}{2\delta_n}\right) = \Delta(n)\Delta(n)^T\nabla V_{\theta(n)}(s_n) + o(\delta_n).$$

The zero order and second order terms directly cancel above instead of being zero-mean, thereby resulting in lower gradient estimator bias. The rest follows from properties (i)-(iii) mentioned previously for the one-simulation gradient SF. In particular, $E[\Delta(n)\Delta(n)^T] = I$.

**Proof of Theorem 2:**

In the light of Proposition 2, the proof here follows in a similar manner as one-simulation SF.

## A.3 Convergence of Two-Simulation Signed SF REINFORCE

Recall that we have

$$H_i(\theta(n), \Delta(n)) = \Delta_i(n)\left[\frac{V_{\theta(n)+\delta(n)\Delta(n)} - V_{\theta(n)-\delta(n)\Delta(n))}}{2\delta(n)}\right].$$

As explained previously,

$$E[H_i(\theta(n), \Delta(n)|\mathcal{F}_n] = \nabla_i V_{\theta(n)} + o(\delta(n)).$$

Also, recall the 'error' in the $i$th component is given by

$$e_i(n) = H_i(\theta(n), \Delta(n)) - \nabla_i V_{\theta(n)} = \sum_{j \ne i} \Delta_i(n)\Delta_j(n)\nabla_j V_{\theta(n)} + o(\delta(n)).$$

**Proof of Theorem 3:**

We rewrite the algorithm as follows:

$$\theta_i(n+1) = \Gamma_i(\theta_i(n) - a(n)sgn(H_i(\theta(n), \Delta(n))))$$

$$= \Gamma_i(\theta_i(n) - a(n)(I(H_i(\theta(n), \Delta(n)) > 0) - I(H_i(\theta(n), \Delta(n)) \leq 0))),$$

where $I(\cdot)$ is the indicator function. Thus, we have

$$\theta_i(n+1) = \Gamma_i(\theta_i(n) - a(n)(1 - 2I(H_i(\theta(n), \Delta(n)) \leq 0)))$$

$$= \Gamma_i(\theta_i(n) - a(n)(1 - 2P(H_i(\theta(n), \Delta(n)) \leq 0|\mathcal{F}_n) + M_i(n+1))),$$

where

$$M_i(n+1) = 2P(H_i(\theta(n), \Delta(n)) \leq 0|\mathcal{F}_n) - 2I(H_i(\theta(n), \Delta(n)) \leq 0),$$

$$= 2P(e_i(n) \leq -\sum_s \nu(s)\nabla_i V_{\theta(n)}(s)|\mathcal{F}_n) - 2I(e_i(n) \leq -\sum_s \nu(s)\nabla_i V_{\theta(n)}(s))$$

$$= 2P(e_i(n) \leq -\sum_s \nu(s)\nabla_i V_{\theta(n)}(s)|\theta(n)) - 2I(e_i(n) \leq -\sum_s \nu(s)\nabla_i V_{\theta(n)}(s)),$$

by (A1). It is easy to see that $(M_i(n), \mathcal{F}_n), n \geq 0$ is a martingale difference sequence. Since $\sup_n |M_i(n)| \leq 1$, and under (A4), it follows from an application of the martingale convergence theorem that $\sum_{m=0}^{n-1} a(m)M_{m+1}, n \geq 1$ is an almost surely convergent martingale.

It is easy to verify that $W(\theta) = \sum_s \nu(s)V_\theta(s)$ itself is a Lyapunov function for the ODE (13) since

$$\frac{dW(\theta)}{dt} = -\bar{\Gamma}(\langle(1 - 2F(-\sum_s \nu(s)\nabla V_\theta(s)|\theta)), \sum_s \nu(s)\nabla V_\theta(s)\rangle)$$

$$= -\sum_{i=1}^N \bar{\Gamma}_i((1 - 2F_i(-\sum_s \nu(s)\nabla_i V_\theta(s)|\theta)) \sum_s \nu(s)\nabla_i V_\theta(s)).$$

From (A3), if $\sum_s \nu(s)\nabla_i V_\theta(s) > 0$, $(1 - 2F_i(-\sum_s \nu(s)\nabla_i V_\theta(s)|\theta)) \geq 0$ and $\frac{dW(\theta)}{dt} \leq 0$. Similarly, if $\nabla_i V_\theta < 0$, $(1 - 2F_i(-\nabla_i V_\theta|\theta)) \leq 0$ and $\frac{dV_\theta}{dt} \leq 0$. Further, when $\sum_s \nu(s)\nabla_i V_\theta(s) = 0$, $\frac{dW(\theta)}{dt} = 0$. From Lasalle's invariance theorem in conjunction with Theorem 2 of Chapter 2 of Borkar (2022), it follows that $\theta(n), n \geq 0$ converges almost surely to the largest invariant set $\bar{K} \subset K \subset \{\theta|\bar{\Gamma}(\langle(1 - 2F(-\sum_s \nu(s)\nabla V_\theta(s)|\theta)), \sum_s \nu(s)\nabla V_\theta(s)\rangle) = 0\}$. The claim follows.

# B  Numerical Results

## B.1  The Stochastic Grid-World Environment

We conduct experiments on a 2-dimensional grid-world environment with configurable sizes denoted by $L$. The agent starts at the top-left corner of the grid-world and aims to reach the terminal state located at the bottom-right corner. The reward function, based on Manhattan distance, penalizes for not reaching the goal, with a gradual decrease in penalty as the agent approaches the terminal state, as visualized in Figure 7. The penalty here is specified by a negative single-stage reward that is high for states that are farther from the goal than the nearby states. This is designed so as to provide guidance to the agent to move in the desired direction towards the goal state.

The agent is allowed to move in all four directions in the grid. However, we introduce an uncertainty in the environment. When an agent executes any of these actions, there is a probability $1 - p$ that the agent will move in the intended direction, and with probability $p/2$, it will move in one of the two perpendicular

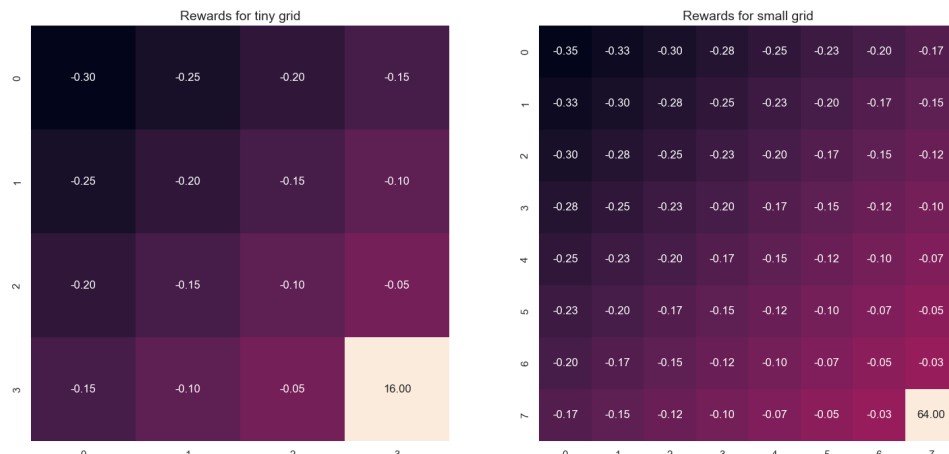

Figure 7: Stochastic grid world problem

| Grid size | Max episode length |
|-----------|--------------------|
| $4 \times 4$ | 50 |
| $8 \times 8$ | 80 |
| $10 \times 10$ | 100 |
| $20 \times 20$ | 150 |
| $50 \times 50$ | 200 |

Table 7: Maximum steps allowed in an episode for a given grid-size (L)

directions. This probabilistic element makes the objective function stochastic. We set $p = 0.1$ in all our experiments. The episode ends after the agent reaches the terminal state. If the agent is unable to reach the goal state, to ensure finite total rewards, we terminate the episode after a specified number of steps. Table 7 shows these specific parameters for the available grid-sizes. Note, however, that only the goal state has a positive reward while all the other states carry a negative (though varying) reward. Thus, only episodes that carry a negative return are the ones that were terminated.

## B.2   Policy function

The input to the policy consists of polynomial and modulo features of the agent's position in the grid. Both algorithms utilize the same policy function, which computes logits for the available actions using a linear function. The logits are then inputted into the Softmax or Boltzmann function to obtain a probability distribution. Figure 8 illustrates the same.

## B.3   Algorithm Parameters

For the two SF-REINFORCE algorithms we chose the sensitivity and step-size parameters as $\delta(n) = \delta_0 (\frac{1}{50000+n})^d$ and $\alpha(n) = \frac{\alpha_0 * 50000}{50000+n}$, respectively, where $n$ is the iteration or episode number. Note that the iteration and episode numbers are the same in our setup. We set $\alpha(0) = \alpha_0 = 2 \times 10^{-6}$. We require $d < 0.5$ to prove convergence theoretically to satisfy the requirement that $\sum_n \left( \frac{a(n)}{\delta(n)} \right)^2 < \infty$. Hence, we restrict these quantities as such. We experiment with both (a) decay schemes by varying $d$ and setting $\delta(0) = \delta_0 = 1$ and also (b) a constant scheme with $d = 0$ and varying $\delta_0$. To reduce the variance, we measure $G_n$ as the average over 10 trials. The two-sided version uses 5 trials for each side, so as to match the net computational effort in comparison to the one-sided scheme, and thereby allowing for a fair comparison.

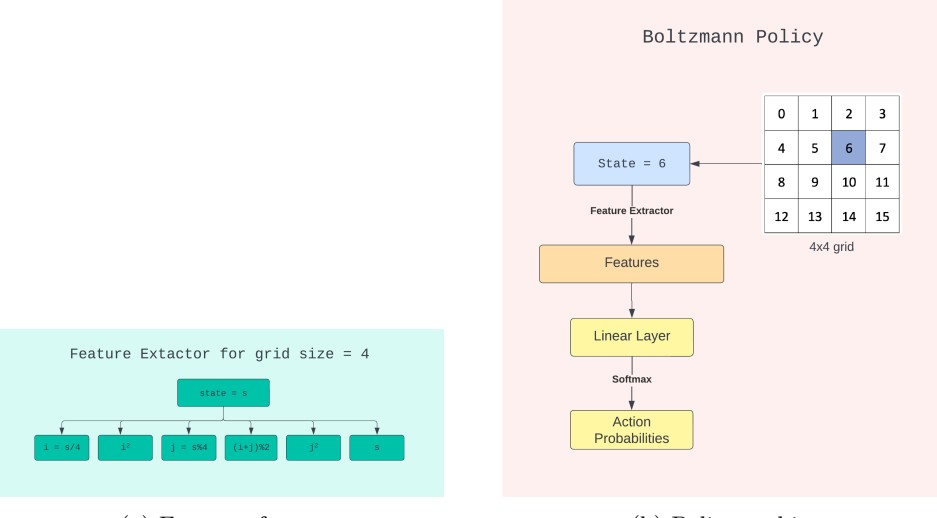

(a) Features from state  (b) Policy architecture

Figure 8: Boltzmann Policy used across all algorithms

| Algorithm / Gridsize | 4x4 | 8x8 | 10x10 | 20x20 | 50x50 |
|---|---|---|---|---|---|
| REINFORCE | $13.90 \pm 0.1$ | $50.74 \pm 2.3$ | $76.90 \pm 4.9$ | $144.89 \pm 113.8$ | $-18.79 \pm 8.4$ |
| PPO | $12.72 \pm 0.2$ | $49.72 \pm 0.2$ | $75.38 \pm 0.3$ | $235.09 \pm 1.9$ | $682.97 \pm 12.9$ |
| SFR-1 | $13.23 \pm 0.1$ | $45.40 \pm 17.7$ | $59.80 \pm 35.0$ | $220.24 \pm 78.8$ | $376.29 \pm 401.4$ |
| SFR-2 | $13.26 \pm 0.1$ | $45.52 \pm 17.8$ | $60.39 \pm 35.0$ | $170.68 \pm 127.4$ | $562.89 \pm 387.7$ |

Table 8: Total reward (mean $\pm$ standard error) of algorithms on different grid sizes. Parameters used: $\delta_0 = 1$, $d = 0.25$ and $\alpha_0 = 2 \times 10^{-6}$

We also include the Proximal Policy Optimization (PPO) algorithm from Schulman et al. (2017b); Raffin et al. (2021) for our numerical comparison. We use the same policy architecture along with a linear layer for the value function. For REINFORCE, and PPO the policy is updated using the policy gradient via an ADAM optimizer with its learning rate set to 0.0003.

## B.4 Gridsizes

Figure 9 illustrates the dynamics of the REINFORCE, SF-REINFORCE and PPO algorithms on various gridsizes. For this, we set the parameters: $\delta_0 = 1$, $d = 0.25$ and $\alpha_0 = 2 \times 10^{-6}$, and vary the gridsize $L = 4$, 8, 10, 20 and 50. For the largest grid-size, we end up having $|S| = 2500$ states. We limit $L \leq 50$ since this is quite a lot for a Boltzmann policy of 30 trainable parameters. Table 8 shows the convergence values.

From Table 8, it can be seen that REINFORCE does only slightly better than our algorithms on smaller gridsizes $L \in 4, 8, 10$ and performs decently for $L = 20$. For smaller gridsizes, since REINFORCE uses the analytical gradient, there is no noise in the gradient due to perturbations unlike SF-REINFORCE. As a result, as seen in figures 9a, 9c and 9d, it is able to quickly converge to the optimal linear policy.

PPO regularizes the objective function by clipping it. Such methods are reliable for policies with deeper neural networks. Yet, we note that PPO is able to work with linear policy and produces consistent performances across seeds.

Compared to REINFORCE, our methods are more robust since it shows reasonable performance on all grid sizes, especially for higher $L \in \{20, 50\}$. The two-sided SF REINFORCE (SFR-2) attains the best performance for $L = 50$. As expected, the one-sided SF REINFORCE (SFR-1) shows similar or higher variance across seeds than SFR-2.

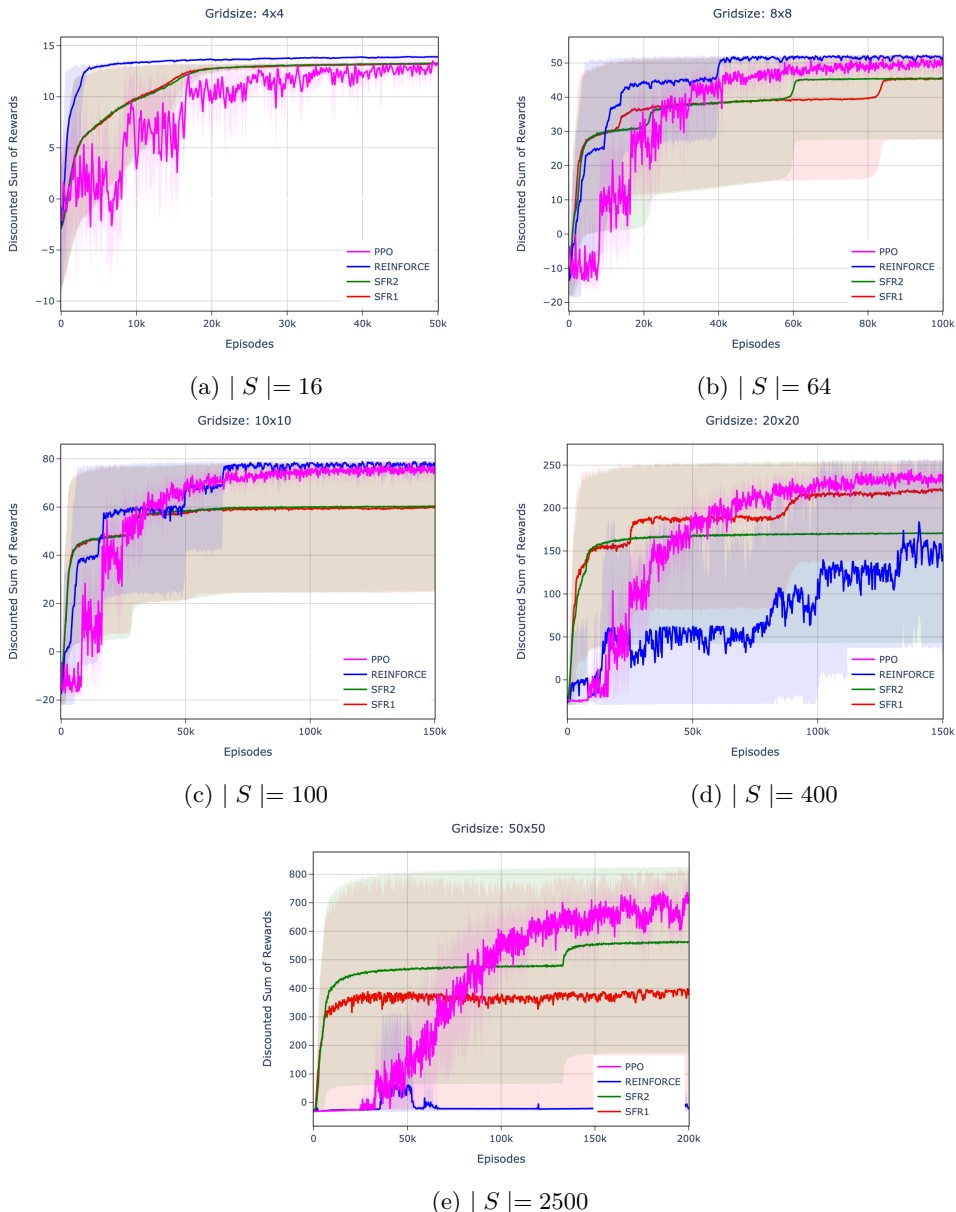

Figure 9: Plots showing performance of iterates of algorithms on various gridsizes. Parameters used: $\delta_0 = 1$, $d = 0.25$ and $\alpha_0 = 2 \times 10^{-6}$

Although our algorithms converge to competent averages for bigger grid sizes, the variance is still quite large. This motivates us to experiment with various perturbation schedules and gradient clipping, and study their effects not only on the performance (mean $\pm$ standard error) but also the dynamics of iterates in Sections 6.1 and 6.2.

