# OpenReview forum: "Variance Reduced Smoothed Functional REINFORCE Policy Gradient Algorithms"
_TMLR — Rejected by TMLR_

### Review · Reviewer_T67W · 2024-05-19

**Summary Of Contributions:**

The paper introduces a functional REINFORCE policy gradient algorithm as alternative to the regular REINFORCE algorithm employed in RL. The paper introduces a few variants, such as the 2-sample version for variance reduction. The paper shows both some mathematical properties of the estimator and carries out experiments to show the method's performance gain over baseline REINFORCE.

**Audience:**

Yes

**Claims And Evidence:**

No

**Requested Changes:**

=== **connections to evolutionary algorithms** ===

Most importantly, how does the method here differ fundamentally from the evolutionary algorithms (ES) proposed in [1] and all the ensuing developments from that paper. Despite the naming of ES, it is actually fundamentally a finite-difference and perturbation based gradient estimation method inspired from blackbox optimization. It is based on trajectory based perturbation rather than action level perturbation such as PPO or REINFORCE. On a high level, the mathematical form and intuition of the functional gradient estimator proposed in this work bears so much similarity with ES, yet there is little to no discussion on the connection to ES in the related work discussion.

It is important to note that this line of work has been much developed in the past few years and there has been lots of variants and improvements on top (e.g., [2]) in high-d continuous controls. Comparing the proposal in this work to such prior work, it feels that the methodologies are mostly similar in mathematical form, and the experimental ablations are quite weak. Maybe the authors can clarify more on the differences between the proposal in this work and such prior work, and carry out proper ablations.

[1] Talisman et al, 2017, Evolution Strategies as a Scalable Alternative to Reinforcement Learning

[2] Mania et al, 2018, Simple random search of static linear policies is competitive for reinforcement learning

=== **1-sample vs. 2-sample** ===

The 1-sample version of the algorithm has too much variance that it can barely work at all in practice. This is intuitively true since 1-sample estimate of the gradient, though mathematically unbiased, has too much variance in it since it cannot reliably estimate a directional quantity.

Hence I am quite surprised about the result in Fig 1 since one-sample estimate without baseline should have very high variance and it is not clear why SFR-1 would work and seem better than REINFORCE and PPO. Not sure if this has to do with the fact that the Fig 1 reports "max" over runs, instead of average.

=== **Fig 1** ===

Fig 1 shows the best run while in RL ablations it is more common to show average runs with standard errors. The current plot basically shows that SFR-2 can achieve better peak performance but does not rule out the possibility that it achieves lower average performance or has much higher variance in practice.

=== **experimental ablations** ===

The paper has carried out ablations around various hyper-parameter designs of the SFR algorithm but didn't compare with more baselines from prior literature, such as ES or AIS [1,2], which are very simple to implement derivative-free gradient estimator similar to what's in this work.

The breadth of the benchmark tasks for evaluation should improve too. So far the experimental results focus on toy grid world domain and it would be sensible to extend them to less toy domains, such as continuous controls as commonly done in RL papers.

**Strengths And Weaknesses:**

The paper is interesting in that it considers functional level perturbation (trajectory level perturbation) to derive gradient estimator. This is an alternative approach to REINFORCE, which considers action level perturbations.

The paper has a few aspects of weaknesses: how does the functional estimator compare to and differ from the evolutionary algorithm style gradient estimator proposed (and has been popular) in the past few years? There is little discussion in the paper about the connection and these two streams of work share lots of common features. The baseline comparison is also quite weak and does not test on high-d challenging domains, while mostly the results are from the toy benchmarks.

---

> ### Author Response · Authors · 2025-04-16
> **Comments of Reviewer T67W addressed in a revision**
>
> 1.We have now given detailed empirical comparisons with the augmented random search
> (ARS) algorithms presented in the literature as also mentioned by the reviewer. We observe that
> these algorithms build on evolutionary strategies (ES) based approaches. We also provide a proof
> of convergence of the ES procedures. Detailed experimental comparisons on the MuJoCo locomotion
> tasks (Hopper, Walker2d, HalfCheetah, and Swimmer) with high-dimensional state-action spaces have
> now been provided.
>
> 2. In the revised version we discuss the evolutionary
> and augmented random search algorithms and discuss some of the relevant literature including the
> papers mentioned by the reviewer. We also provide a proof of asymptotic convergence of the ES
> evolutionary algorithms that the previous literature was lacking as some of the prior work mainly
> focused on providing finite time sample complexity bounds on the ES algorithms. The augmented
> random search (ARS) algorithms build on ES and have been shown in prior work to be superior in
> performance. We show performance comparisons now of our algorithms with ARS.
>
> 3. We agree that the 1-sample version does indeed suffer from high variance. Since SFR-2
> is better in performance in most settings over SFR-1, in the revised manuscript, based on the com-
> ments of the reviewers, we mainly show empirical comparisons of SFR-2 with the ARS algorithms
> over high-dimensional MuJoCo environments, and in Table 3, we refer to comparisons with PPO,
> soft actor-critic etc., reported in Mania et al., over these environments. When compared with ARS,
> our results indicate that for a given number of environment interactions, SFR-2 is competitive when
> compared with ARS, in fact better than ARS on half of the environmental settings considered.
>
> 4. We agree with the reviewer. We now show the performance of the various algorithms
> by taking the mean and standard error across various seeds in the main figures and tables shown in
> the paper. We also show in Appendix B.2, the plots with the best seeded runs for each algorithm for
> more clarity.
>
> 5. (Experimental Ablations): We agree with the reviewer and have now included the results of detailed experiments
> on four different MuJoCo environments namely Hopper, Walker2d, HalfCheetah, and Swimmer, that
> involve high-dimensional state-action spaces. In all of these environments, we now compare the perfor-
> mance of our algorithms (including the signed and clipped gradient versions) primarily with ARS-v1t
> and ARS-v2t, respectively. We observe from our experimental results that our algorithms are com-
> petitive when compared with ARS-v1t and ARS-v2t.

---

### Review · Reviewer_4ymX · 2024-06-23

**Summary Of Contributions:**

This paper revisits the REINFORCE policy gradient algorithm and proposes an enhanced version that estimates the gradient using a smoothed functional gradient estimator with random perturbation (SF-REINFORCE). They also offer a two-simulation estimator with a lower bias (SF2-REINFORCE). Their methods also suffer from a large variance, so they develop two variants: one replaces the increment term in their gradient updates with the sign of the increment; the other one adds clipping to the costs (as is done in PPO).

They theoretically analyze the convergence properties of SF-REINFORCE and SF2-REINFORCE, showing that the parameter vector converges to the interior of the equilibria set. They also did experiments on a grid environment with different grid sizes. They claim that SF-REINFORCE and SF2-REINFORCE outperform REINFORCE and PPO (when comparing their best performance).

**Audience:**

Yes

**Broader Impact Concerns:**

/

**Claims And Evidence:**

Yes

**Requested Changes:**

See above.

**Strengths And Weaknesses:**

Strengths:

1. They propose novel algorithms and improve the classical algorithm REINFORCE. The algorithm is simple and easy to implement, as it only needs a simulation of some Gaussian random vectors.

2. They provide a rigorous convergence analysis for SF-REINFORCE and SF2-REINFORCE. The proof is correct, to my knowledge.

Weaknesses:

1. The paper lacks experiments on a larger dataset and real environment. Since they want to compare the proposed algorithms with REINFORCE and PPO, which are already applied in real scenarios, it is expected that the proposed algorithms have a practical version and can be comparable to them in real scenarios.

2. The experiment results are not very clearly presented. For example, in Table one, what does each column mean? Does 'clip_by_ value' stand for the clipped algorithm with original increments or signed increments? The authors should annotate them well in the caption for each table. There are several figures and tables, and different versions of algorithms are tested, so the authors should make them clearer.

3.  I think the authors should have a more thorough comparison in the main text. For example, in Table 8, the SFR-1 and SFR-2 are outperformed by PPO. What version of SFR-1 and SFR-2 are used to produce this table? Are they with or without signed increments / clipped values? Basically, since there are several different implementations for the SFR-1 and SFR-2, I think it is best to show a full list like Table 9 in the main text to compare all algorithms on all grid sizes.

4. In the intro and algorithm sections, the authors emphasized that one condition that the classical policy gradient method and REINFORCE assume is the interchangeability of the expectation and the gradient, which is not always satisfied. Although this is true mathematically, the authors did not show in which case this can fail, either by theory or by experiments. Personally, I have not read any literature about the interchangeability of expectation and gradient in the policy gradient theorem, and this is often assumed to be true in most cases. I do not think this is a very good motivation for the proposed algorithm and it can rather distract readers, so I personally suggest to remove this argument as this is not the main issue that the authors aim to solve.

---

> ### Author Response · Authors · 2025-04-16
> **Comments of Reviewer 4ymX addressed in a revision**
>
> 1.Based on the reviewer’s comments, we now focus our attention on large scale MDPs and
> show the results of experiments on MuJoCo locomotion tasks, namely, Swimmer, Hopper, Walker2d
> and HalfCheetah, respectively, involving high-dimensional state spaces. Based on the comments of
> the other reviewers, we now compare our results here with Augmented Random Search algorithms,
> see Mania et al (2018). The latter reference also shows the comparisons with other algorithms such as
> A2C, PPO, TRPO and CEM. We recall those results and further show the comparisons of our results
> and those of ARS. Since we are dealing with online learning, we ensure a fixed number of interactions
> with the environment across all algorithms. Under this setup, we note that our algorithm is competitive against other algorithms and shows good performance, better than ARS on many occasions.
>
> 2. We have now simplified the experimental comparisons. Since SFR-2 is seen to be better
> than SFR-1, we only now show performance comparisons with SFR-2. For the same reasons, we do not consider the one function sample estimators of the gradient considered in Salimans et al., and focus
> our attention only on ARS algorithms which use two measurement estimators alone as they are better
> than their one-simulation counterparts. We have also improved the organization of the experimental
> comparisons to make them more focused.
>
> 3. Over the new set of experiments on high-dimensional domains, we indeed now show
> in Table 2, the experimental comparisons with the state-of-the-art algorithms ARS-v1t and ARS-v2t.
> We compare the various variants of SFR-2 including component-clip, norm-clip and signed updates,
> with corresponding variants of both ARS-v1t and ARS-v2t, respectively. We observe that our algo-
> rithm is consistently better on all the aforementioned variants than both ARS-v1t and ARS-v2t, on
> the Swimmer and HalfCheetah environments. However, as expected, the original SFR-2 algorithm
> does not show as good results as the original ARS-v1t and ARS-v2t (i.e., without clipping and signed
> updates). On Walker2d, SFR-2 with Component Clip is better than ARS-v2t but is not as good
> as ARS-v1t on this task. This goes in to show that despite the fact that we use only two function
> samples per iteration to estimate the gradient, unlike ARS that uses 2k (and then selects the best b
> directions) for an a priori chosen k > 1, our algorithms for a fixed number of environment interactions
> are competitive against ARS.
>
> 4.We agree with the reviewer and have now removed all discussion pertaining to this
> interchange of operators. Also algorithms such as evolutionary strategies (ES) and augmented ran-
> dom search (ARS) involve zeroth order methods that we compare our algorithms with. As an aside,
> we also now prove the asymptotic convergence of the ARS algorithms that had not been shown in
> earlier papers. In Mania et al. (2018), numerical comparisons of ARS with the other algorithms such
> as PPO, TRPO, CEM, and A2C has been shown and it is observed that ARS is often times superior
> to these other algorithms. We thus compare SFR-2 with the ARS algorithms and recall the other
> comparisons already obtained by Mania et al in Table 3.

---

### Review · Reviewer_FYUY · 2024-08-16

**Summary Of Contributions:**

This paper studies variations of Reinforce. These variations are based on what seems to be a classic approach in zero-order stochastic optimization, that is (even if not stated that way), the idea is to follow the gradient of a Gaussian-blurred version of the objective function. Four variations are considered (a single sample or a two-sample exploiting a symmetrized blurred objective, applying or not the sign to the descent direction). An asymptotic convergence analysis is provided, as well as an empirical study on small grid-worlds.

**Audience:**

Yes

**Claims And Evidence:**

No

**Requested Changes:**

Please address the above comments.

**Strengths And Weaknesses:**

The proposed algorithms are interesting, but at least half of them are not that new (both from an empirical and a theoretical perspective), the authors miss a relevant part of the literature. There are also a number of clarity issues as well as some unsupported claims. Details below.

## Position to previous works

(Signed) SFR 1 and 2 are zero-order (or black-box) optimization approaches to reinforcement learning. This is not said in the paper, but there is a large literature on the subject that should be discussed. The following two papers are especially relevant, but the list is probably not exhaustive.

**“Evolution Strategies as a Scalable Alternative to Reinforcement Learning”, Salimans et al, 2017**:
This is an empirical paper, and the “evolution strategy” considered is exactly SFR-1. They apply it to teach deep neural networks to play Atari.

**“Derivative-Free Methods for Policy Optimization: Guarantees for Linear Quadratic Systems”, Malik et al, 2020**:
This is a theoretical paper that analyzes both SFR-1 and SFR-2 (see Eqs. 12 a-b). They study the more restricted class of LQ systems, but they provide a non-asymptotic convergence guarantee. It seems necessary to position the presented theoretical results to this work, and ideally explains where the analysis differs, what allows to study a more general case (at the cost of not necessarily converging to the optimal solution, without rate of convergence). It should be also acknowledged that these algorithms are not new.


These are only two examples, but given the number of times both papers have been cited, there may be other relevant works. It would be also quite interesting to compare your convergence results to the one of Reinforce, which is now quite well studied. For example, see **“A general sample complexity analysis of vanilla policy gradient”, Yuan et al (2020)**, as well as the references therein, for a good overview.



## Unsupported claims

The title starts with “Variance reduced”, but the variance is not studied theoretically, and the empirical evidence is not really convincing.  There are other claims not being supported. For example, it is stated in the introduction (about SFR-1)
>"The problem, however, is that it suffers from a large bias in the gradient estimator. We show analytically the reason for the large bias here."

This is wrong, no such analysis is provided. Still in the introduction, there’s a discussion about
>“interchange between the expectation and gradient operators [...] not easy to justify this interchange when the number of states is infinite resulting in an infinite summation or an integral when computing the expectation.”

However, only finite state and action spaces are considered in the paper (so it is unclear if it alleviates the discussed issue, which is by the way not very clear). In section 4, it is stated
>“The first variant reduces the bias in the estimator by using two simulations instead of one, while the second variant uses the sign of the increments instead of the increments themselves and this helps reduces the estimator variance.”

So the initial bias of the estimator (SFR-1 from the context) is never provided and even less discussed. There’s no evidence that SFT-2 reduces the bias (and using more samples lets rather think of reducing the variance than the bias). Same for using the sign, why would this reduce the variance, what are the evidences?

## Clarity issues

The paper has a number of clarity issues.

First, for being didactic, it would be really great to explain how the gradient estimators are obtained, it would really help widening the audience. For example, if one want to optimize for $F(\theta)$, one could consider a Gaussian-blurred variation $E_{\epsilon\sim\mathcal{N}(0,I)}[F(\theta + \sigma \epsilon)]$, the gradient becomes $\frac{1}{\sigma}E_{\epsilon\sim\mathcal{N}(0,I)}[F(\theta + \sigma \epsilon) \epsilon]$ (score function estimator), which gives the considered empirical gradient for SFR-1. The blurred objective $E_{\epsilon\sim\mathcal{N}(0,I)}[F(\theta + \sigma \epsilon) + F(\theta - \sigma \epsilon)]$ gives the gradient of SFR-2. It is also a problem that the whole paper is discussing Reinforce, while never defining the reinforce algorithm.

The policy gradient in sec. 2.2 is suspicous. To the best of reviewer’s knowldege, this policy gradient was derived by Sutton et al for the average case and the discounted case, not for shortest path problems (SPP). For an SPP, given the definition of $\eta$, the whole probability mass would be on the terminal state, and thus $\eta(t) = 1$ with $t$ the terminal state. It does not really make sense.

There are various notation issues. For example, $a$ is used for actions, for learning rate and for defining the set $C$ onto which one projects. The notation $H$ is also used for different things, as well as the notation $Z$.
The reviewer did not carefully checked the proofs, but they could be more clearly presented. They probably build upon classic analysis, it would be worth discussing what are the key steps making it differ from what is done for analyzing this kind of zero-order optimization approach more generically. It would be also helpful to state clearly the assumptions when stating the results, not lost in multiple parts of the text.

## Experimental study

Only very small scale MDPs are considered. The information is very informative, basically the optimal policy is encoded within the reward function (a myopic RL algorithm, that is with $\gamma=0$, would find the shortest path). The proposed approach is claimed to work better than PPO or Reinforce, but nothing let think that the baselines have been tuned as much as the proposed approaches (only a single HP, the learning rate, is mentioned). It is stated in appx B.4 *“since Reinforce uses the analytical gradient”*, which is really surprising. What do the authors call reinforce?

Another issue is the motivation for gradient clipping. Yes, it makes sense, there's some weak relationship to the signed update (that could be at least briefly discussed), but this is a general approach that would apply to PPO, Reinforce, or whatever is gradient-based (and so the considered baselines should also be tried with gradient clipping). In the end, it also seems that gradient clipping is more  beneficial than the signed update, while it is the only new algorithm (to the best of reviewer knowledge).

---

> ### Author Response · Authors · 2025-04-16
> **Comments of Reviewer FYUY addressed in a revision**
>
> 1. We have now given references to these zeroth order
> optimization approaches that have been recently studied. We also provide qualitative comparisons of
> our algorithms with these methods, as well as an asymptotic analysis of convergence of ES approaches
> (in addition to our methods) – something that was lacking in the earlier literature on ES approaches.
> We have also now provided detailed experimental comparisons of our methods (including those with
> clipping and signed updates) with similar versions of the ARS algorithm. Our objective is stochastic, nonlinear, nonconvex (in general) and thus can possess any number of local optima. Regarding comparisons with Yuan et al., we note first
> that Yuan et al., treat the discounted reward case, not the stochastic shortest path problem. Moreover,
> they also make several strong assumptions on smoothness of the objective, its truncation as well as
> assumptions such as on weak gradient domination, and provide a sample complexity (finite time) anal-
> ysis but do not provide asymptotic convergence guarantees like we do.
>
> 2. We have now shown in Lemma 1 that SFR-2 has lower gradient estimation bias than
> SFR-1. Further, in Lemma 2, we have shown the following: (i) the variance of the gradient estimators
> with the signed updates is uniformly upper bounded by 1. Such a uniform bound cannot be provided
> for the variance of the regular (unsigned) updates because of the form of the gradient estimators
> that involves terms containing the sum of rewards in the numerator and a small quantity δn in the
> denominator, see Lemma 2(i) and Remark 1(i).
> In Lemma 2(ii), we show that the variance of the gradient estimators with clipping is less than
> the regular estimators in SFR-1 and SFR-2 because the clipping function in either case of f = fc and
> f = fm is a projection that we further show is a non-expansive map. We further observe variance
> reduction empirically in both cases over the regular gradient estimators (see Tables 2 and 3).
> We have also now removed the discussion about the interchange between the gradient and expectation operators.
>
> 3. The proof of the policy gradient theorem in the episodic case (that we consider) is given in the
> RL book of Sutton and Barto, 2018, see page 325 of the book (cited in the paper) for a proof of the
> theorem. The way to analyze the episodic or stochastic shortest path case is to generate multiple
> trajectories one after the other where the start time of a trajectory (if it is not the first) comes the
> instant after the end time of the previous trajectory where the start state in each trajectory is chosen
> randomly as per the given initial distribution, see Sutton and Barto (2018).
>
> 4.  We have highlighted that the ES and ARS papers make several strong assumptions on the system
> model and the objective function which are often not practically realizable, unlike us. We show the
> asymptotic convergence analysis of our algorithms under very general assumptions. In fact, all our
> results are shown under two assumptions – these are listed in the paper as Assumptions 1 and 2,
> respectively, in sections 2 and 3. We in fact show the various requirements needed.
>
> 5.We now focus our attention on large scale MDPs. We show the results of experiments on
> MuJoCo locomotion tasks, namely, Swimmer, Hopper, Walker2d and HalfCheetah, respectively, in-
> volving high-dimensional state spaces. We compare our results here with Augmented Random Search
> algorithms, see Mania et al (2018). The latter reference also shows the comparisons with other algo-
> rithms such as A2C, PPO, TRPO and CEM. We recall those results in Table 3 and further show the
> comparisons of our results and those of ARS. Since we are dealing with online learning, we ensure a
> fixed number of interactions with the environment across all algorithms. Under this setup, we note
> that our algorithm is competitive against other algorithms and shows good performance, better than
> ARS about half the number of times.
>
> 6.We show in Table 2 the performance comparisons now of our algorithm with ARS-v1t
> and ARS-v2t both in their original versions and also with gradient clipping (both component-wise
> and norm-clip) and signed updates, where these variants have also been tried on the ARS algorithms.
> SFR performs updates more often than ARS, albeit they are bound to have higher variance. This
> motivates us to use clipped and signed gradients as it reduces variance and improves performance. We
> observe that our algorithm is better on all variants than both ARS-v1t and ARS-v2t on the Swimmer
> and HalfCheetah environments, though the original SFR-2 algorithm does not show as good results
> as the original ARS-v1t and ARS-v2t (i.e., without clipping and signed updates), see Table 2. On
> Walker2d, SFR-2 with Component Clip is better than ARS-v2t but is not as good as ARS-v1t on this
> task. This goes in to show that SFR-2 with just two environment interactions per parameter update
> (as against 2k for SFR-2) is competitive against both ARS-v1t and ARS-v2t.

---

### Comment · Action_Editor_NA9r · 2024-07-02
**Pardon delay**

Dear authors,
This is just a note apologizing for the delay in the review of your manuscript. We've had some issues finding reviewers accepting to review, as well as reviewers accepting but not submitting their reviews.

We are on it, and hope it won't be delayed too much longer!

---

> ### Comment · Action_Editor_NA9r · 2024-08-14
>
> Just to follow up here: I'm still having a hard time finding a third reviewer but am actively looking. I'm hopeful this will be resolved in the next few weeks.
>
> Once again, apologies for the massive delay!

---

### Decision · Action_Editor_NA9r · 2024-10-07

**Recommendation:** Reject

**Comment:**

The reviewers were unanimous in suggesting that this paper is not yet ready for publication, which I agree with. Given that the authors did not provide a single response to any of the reviews suggests they agree.

I would encourage the authors to provide a more thorough empirical evaluation and literature review, and perhaps revisit some of their claims based on these findings.

**Audience:**

Yes.

**Claims And Evidence:**

As pointed out by all reviewers, this paper is currently lacking in empirical evaluations of their findings. Specifically, they evaluate only on small-scale MDPs, and on these do not provide sufficient ablations.

Further, some of the theoretical claims (e.g. variance reduction) are not well-supported, and relevant literature was not properly studied (see detailed comments by reviewer FYUY).

**Resubmission Of Major Revision:**

The authors may consider submitting a major revision at a later time.

---

> ### Author Response · Authors · 2025-04-16
> **Revision submitted**
>
> We have now addressed all the concerns of the reviewers through substantial revisions in the revised version of our manuscript that we have now submitted. We briefly indicate the changes made below.
>
> 1. Broader experimental evaluations on MuJoCo tasks: We show the results of detailed experiments
> on MuJoCo locomotion tasks for Swimmer, Hopper, Walker2d and HalfCheetah environments.
> We observe that for the same number of environment interactions, SFR-2 with clipping and
> signed updates performs consistently better over two of the four tasks.
>
> 2. A detailed discussion of related work including evolutionary strategies and augmented random
> search based approaches as well as detailed experimental comparisons (as mentioned above) have
> now been performed.
>
> 3. Clarifications and improvements in theoretical and empirical results have been provided. More
> specifically, we have now given proofs for the bias reduction claim (Lemma 1) and variance reduction claim (Lemma 2 and Remark 1). The detailed proofs of these results are provided in
> Appendix A.3 and Appendix A.4, respectively. Furthermore, we also prove asymptotic convergence of the ES and ARS algorithms mentioned by the reviewers. Prior work had provided only
> finite time (non-asymptotic) analyses for these algorithms that too under much stronger requirements. We prove the asymptotic convergence under just two assumptions, namely Assumptions
> 1 and 2. In fact, we prove all the basic requirements such as the parameterized value function
> being differentiable with a Lipschitz continuous gradient (Lemma 4). This is unlike the papers
> on ES/ARS that make much stronger assumptions but do not prove whether these assumptions
> are valid in the settings that they consider.
>
> 4. We thank the editor for giving us the opportunity to revise the paper and resubmit our work
> for evaluation.

---

> > ### Comment · Action_Editor_NA9r · 2025-04-28
> >
> > Dear authors,
> >
> > As per [the website](https://jmlr.org/tmlr/editorial-policies.html):
> >
> > Authors of a rejected submission may revise and resubmit their paper, but it will need to be entered as a new submission and a link provided to the previously rejected submission as well as a description of the changes made since.

---

> > > ### Author Response · Authors · 2025-04-29
> > > **Resubmission of the paper made as a new submission (#4685)**
> > >
> > > Dear Editor,
> > >
> > > Yes, indeed, we have followed the due process as mentioned on the website. Our new paper submission number is 4685. We have provided a link to this (rejected) earlier submission in the new paper and also given a description of the changes made in that (new) submission. Since there wasn't enough space to provide a point-by-point response to all the reviewers of the earlier submission, we have entered the aforementioned response individually for each reviewer of our earlier submission here (on this page of the earlier paper).
> > >
> > > Regards,
> > > Authors of 2436 (with new submission #4685)